
# Residential building stock modelling for mainland China

Danhua Xin[1], James Edward Daniell[1,2,*], Hing-Ho Tsang[3], Friedemann Wenzel[1]

[1]Center for Disaster Management and Risk Reduction Technology (CEDIM) and Geophysical Institute, Karlsruhe Institute of Technology, Hertzstrasse 16, 76187, Karlsruhe, Germany

[2]General Sir John Monash Scholar, The General Sir John Monash Foundation, Level 5, 30 Collins Street, Melbourne,Victoria, 3000, Australia

[3]Centre for Sustainable Infrastructure, Swinburne University of Technology, Melbourne, VIC 3122, Australia

*Correspondence to James Edward Daniell (j.e.daniell@gmail.com)

## Abstract

Previous seismic damage reports have shown that the damage and collapse of buildings is the leading cause of fatality and property loss, especially in developing countries. To better serve the risk analysis targeted at near-real-time post-earthquake mitigation and pre-earthquake preparedness and resources allocation, this study develops a fully reproducible grid-level residential building stock model for mainland China, by disaggregating urbanity level census data of each province into 1km×1km scale and using population density profile as the proxy. To evaluate

the model performance, the modelled residential building stock value is compared with the net capital stock value in Wu et al. (2014) using perpetual inventory method at provincial level. The modelled stock values in these two studies are in good agreement for all the 31 provinces in mainland China. Furthermore, district level comparison of the residential floor area developed in this study with records from statistical yearbook of Shanghai is also conducted. It turns out that the floor area developed in this study is compatible with floor area recorded in the

yearbook of Shanghai. To further validate the applicability of the modelled results in seismic risk assessment, an estimation of the scenario loss to modelled residential buildings is performed, by assuming the recurrence of 2008 Wenchuan M8.0 earthquake. The overall estimated loss approximates the loss value derived from damage reports based on field investigation quite well. Both results indicate the reliability of the residential building stock model developed in this study. The limitations of this study are discussed and directions for future work are recommended.

## 25   1.   Introduction

With the theme of last year's International Day for Disaster Reduction (IDDR2018) being "Target B: Reducing the number of affected people by disasters by 2030", the awareness of the impacts of natural disasters on human society has been increasing over the years. Demands from public sector for quantification of disaster risk is thus more urgent than before. As stated by António Guterres, the current United Nations Secretary-General, in

IDDR2018, that "Disasters cost hundreds of billions of dollars (every year), hitting the poorest countries disproportionately and pushing millions into poverty. We must tackle disaster risks and leave a more resilient planet to future generations." To better cope with the frequent occurrence of earthquakes and other natural hazards (typhoon, flood, tsunami, etc.), the development of sound risk models for natural hazards should be given top priority, since these hazards can lead to tremendous and often crippling economic losses especially in the countries


of the developing world. According to the estimation in Daniell et al. (2011, 2017), from 1900-2016, 2.3 million earthquake fatalities from 2233 fatal events occurred worldwide, with economic losses (direct and indirect) associated with the occurrence of over 9,900 damaging earthquakes reached USD 3.41 trillion (in 2016 price level).

To develop a seismic risk model, three layers of information are essential: hazard, exposure and vulnerability. Hazard refers to the occurrence frequency and severity of ground shakings generated by earthquakes. Exposure
captures the attributes of exposed elements in terms of value, location and relative importance (e.g. buildings, critical facilities and infrastructure) to potential earthquake. Vulnerability describes the susceptibility of those exposed elements to earthquake. Among the exposed elements, buildings are considered as the most important asset category in seismic risk assessment, since the majority of loss and fatality that occurs during earthquakes are related to building damage and collapse (Neumayer and Barthel, 2011; Yuan, 2008). As such, estimation of the
building stock and the values at risk is an important and integral part of any risk modeling effort. Specifically in developing and disaster vulnerable countries like China, rapid urbanization process has led to massive increase in both the asset value and population exposed to seismic hazards (Hu et al., 2010; Yang and Kohler, 2008). Therefore, a country-level modelling of the building stock and its spatial distribution across China is essential.

Ideally, if the building stock value of the research portfolio is already known, e.g. in an insurance portfolio,
building attributes (i.e. the location, geometry, height, construction age and material, occupancy type etc.) are used mainly for building vulnerability determination. However, in most cases, the building stock value is not available and obtaining such detailed information for every building in a large region is not practicable. Therefore, the aforementioned building attributes, which are usually provided at administrative level in census data, are also used to estimate the building stock value. In this case, appropriate proxy (e.g. population density) is required to
disaggregate administrative level census data into finer scale. The use of proxy is quite a reasonable approach in dasymetric modelling and has been frequently adopted in previous studies (e.g. Gunasekera et al., 2015; Silva et al., 2015; Thieken et al., 2008).

When disaggregating census data into a finer scale, it cannot be carried out by simply assuming that the assets within an administrative unit are evenly distributed, since in reality people and buildings tend to be concentrated
in settlements e.g. along the riverside or within alluvial plains (Figueiredo and Martina, 2016). In this regard, more sensible techniques have been applied and documented in the literature. For example, Silva et al. (2015) disaggregated the building stock at parish level for mainland Portugal based on the population density profile at $30 \times 30$ arc-sec resolution cells from LandScan. The LandScan population density profile was produced by apportioning best available census counts into cells based on probability coefficients, which in turn were derived
from road proximity, slope, land cover and night-time lights (Dobson et al., 2000).

In mainland China, the modelling of building stock value and its spatial distribution across China is scarcely done at high-resolution (e.g. 1km×1km scale). In those published studies related to building stock model development, e.g. Yang and Kohler (2008) and Hu et al. (2010), the simulation and evolution of building stock value (taking the mainland China as a whole) were designed and targeted for resource consumption and environmental impacts
purposes, which cannot meet the needs in risk analysis due to their coarse resolution. International projects e.g. PAGER (Jaiswal et al., 2010) and Gunasekera et al. (2015) also conducted global exposure modelling that covered the building stock value in mainland China. However, these global models cannot fully make use of the census




data available in each country and usually assuming a uniform distribution of building stock value per capita for each province or even for each country, which might be convenient, but not realistic, especially for unevenly

developed countries like China. A recent work of Wu et al. (2018) established a high-resolution (1km×1km scale) asset value model based on the net capital stock value they estimated for 344 prefectures in mainland China using the perpetual inventory method (Wu et al., 2014). However, their original asset data to be disaggregated into grid level was actually restricted to prefecture level. Furthermore, the extent of the natural hazards, in most cases, are dependent on the geological structure (earthquakes) or along the riverside (floods), instead of being restricted to

administrative boundaries. Therefore, to better cope with this spatial mismatch between natural hazards and administrative boundaries, building stock models should be geo-coded with relatively high resolution and be disaggregated from more detailed census data.

The organization of the following sections is as follows: the full list of data sources needed, and a detailed description of the methodology used to develop the high-resolution building stock for mainland China will be

firstly introduced. Then, to evaluate the model performance, provincial and district level comparison of the modelled results with that in previous studies and yearbook records will be conducted. Finally, an application of the building stock model in seismic risk analysis will also be given.

## 2.    Data Sources and Methodology

This section will introduce in detail the building related census data needed to develop the building stock model

and the methodology used to disaggregate the administrative level census data into grid level. The census data used in this study for building stock modelling are extracted from the Tabulation of the 2010 Population Census of the People's Republic of China (hereafter abbreviated as the "2010-census"), particularly for residential buildings. Like in most countries of the world, the national level population and housing census are carried out at 10-year interval, and currently the latest version was issued in 2010. In the 2010-census, there are two types of

tables: Long Table and Short Table. Long Table includes summaries based on the surveys of 10% of the total population in mainland China, while the Short Table summaries are based on the surveys of the whole population. Building stock model related census data (e.g. building occupancy type, height classes, construction material, etc.) are extracted from the Long Table of the 2010-census. Supplementary demographic information (e.g. the total population, the average number of people per family and average floor area per person) are extracted from the

Short Table of the 2010-census. The data of the 2010-census are summarized in Table 1.

In the 2010-census, for each of the 31 provinces, autonomous regions and municipalities in mainland China (hereafter, all referred to as provinces), the building related census data in the Long Table are categorized into three urbanity levels (urban, township and rural), based on the administrative belonging of the surveyed population. The building related census data for each urbanity level of each province are listed in Table 2. Compared with

provincial level census data used in previous studies, one advantage of the 2010-census data is its further categorization of data into three urbanity levels, which better reflects the regional difference within each province.

To disaggregate the urbanity-level based census data into grid-level, population density is used as the proxy, as is a common practice in risk analysis (Aubrecht et al., 2013). The population density profile chosen in this study is developed by Global Human Settlement (GHS) project of the European Commission in 2015, which was



disaggregated from census or administrative units to geo-girds, informed by the distribution and density of built-up area as mapped in their Global Human Settlement Layer (it is worth noting that this dataset has been updated in 2019). In the 2015 GHS population density profile, the number of population in each geo-grid is given. When compared to values from population counts they prove to be accurate (Gunasekera et al., 2015). The original resolution of the 2015 GHS population density profile is 250m×250m, for calculation convenience it is resampled

to 1km×1km resolution before further analysis. The provincial boundary (level 1) vector layer dataset defining the spatial boundaries of mainland China is from the Global Administrative Areas (GADM, www.gadm.org).

With these data on population and residential building stock, a top-down spatial scaling method will be performed to disaggregate the urbanity-level census data into 1km×1km resolution grids for each province in mainland China. The flowchart in Fig. 1 provides an overview of this modelling process. Detailed explanations of each component

and step are as follows.

**2.1: Assign urbanity attribute (urban/township/rural) to the geo-coded grids in the 2015 GHS population density profile**

As outlined above, the population and housing related census data for each of the 31 provinces in mainland China are categorized into three urbanity levels: urban, township and rural. Therefore, the geo-coded grids in 2015 GHS

population density profile should also be assigned with an urbanity attribute first, before disaggregating the urbanity-level based census data into each grid. For each province, this is achieved by applying the population reallocation approach developed by Aubrecht et al. (2015) and also illustrated in detail in Gunasekera et al. (2015).

Following this population reallocation approach, the urban/township/rural population proportion of each province can be derived from the Short Table of the 2010-census (as listed in Table 2). For example, in Shanghai City

(which is one of the four municipalities in China), the population proportion of urban/township/rural urbanity level is 76.64%, 12.66% and 10.7%, respectively. Then the grids (1km×1km) in 2015 GHS population density file of Shanghai are sorted from the largest to the smallest, and the population in those largest and most populated geo-codes grids are summed up and selected until the 2010-census urban population share (i.e. **76.64%** for Shanghai) is reached. These selected grids are thus assigned with urbanity attribute "urban". The smallest population of these

selected grids is taken as the threshold to divide urban and non-urban grids (for Shanghai this urban/non-urban population density threshold is **4827** per km$^2$). For the remaining non-urban grids, the same process is repeated iteratively until the township population proportion (i.e. **12.66%** for Shanghai) is reached. These secondly selected grids are assigned with urbanity attribute "township" and the smallest population among these grids is taken as the threshold to divide township and rural grids (for Shanghai this township/rural population threshold is **2736** per

140    km$^2$). The remaining grids are thus assigned with "rural" attribute. The distribution of the assigned urban/township/rural grids in Baoshan District of Shanghai City is shown in Fig. 1 as an example.

Reiterate the above calculations for all the 31 provinces in mainland China, then all the geo-coded grids in the 2015 GHS population profile can be assigned with urban/township/rural attribute accordingly. The corresponding population thresholds for each province are provided in Appendix Table A1.


**2.2: Step 1-Extract the building related census data from the Long Table of the 2010-census (statistics derived from surveys of 10% population of mainland China.**

As in many other countries, the population and housing census data in mainland China are particularly surveyed for residential buildings. Therefore, the building stock model developed in this study is for residential building stock. Related census data for assessment of residential building stock value include the number of families living

in building types grouped by building occupancy (i.e. residential, commercial, mixed), by number of storey (i.e. 1, 2-3, 4-6, 7-9, ≥10), and by construction material (i.e. steel/reinforced-concrete, mixed, brick/wood, other; hereafter steel/reinforced-concrete is abbreviated as steel/RC; and "mixed" refer to different combinations of masonry buildings). As already listed in Table 1, these data are extracted from the Long Table of the 2010-census, based on the survey of 10% of the total population in mainland China. Therefore, to evaluate the whole building

stock value across China, these building related 2010-census data should be extended from 10% to 100% population first by multiplying the factor of 10 (namely factor $F0$ in Step 1-1 of Fig. 1).

After multiplying the factor of 10, the overall number of families living in building types grouped by building storey or construction material is considered to be complete for each urbanity level of each province. With the family number living in each building type known, by multiplying the average number of population per family

(namely factor $F1$ in Step 1-2 of Fig. 1), which is also provided in the Short Table of the 2010-census, the overall population living in building types grouped by storey (1, 2-3, 4-6, 7-9, ≥10) or construction material (steel/RC, mixed, other, brick/wood) can thus be instantly derived for each province and each urbanity level.

Up to now, the geo-coded grids in the 2015 GHS population density profile have been assigned with urbanity attribute and the population living in each building type is also derived for each province and each urbanity level

from the 2010-census. It is noteworthy that the changes in population or building from 2010 to 2015 has not been considered yet. In rapid urbanization countries like China, the bloom of construction of buildings and the population inflow from township/rural areas to urban areas are significant. Therefore, the population derived from the 2010-census needs to be further amplified to the 2015 level, and mathematically this amplification factor (factor $F2$ in Step 1-3 of Fig. 1) is assumed to be equal to the ratio between 2015 GHS population and 2010-census derived

population (after amplified from 10% to 100% of the population).

As listed in the last column in Table 2, the amplification factor $F2$ varies across each urbanity level of each province (namely factor $F2$ in Step 1-3 of Fig. 1). For each province, $F2$ in the urban area is generally higher than in township/rural area, which is quite reasonable. However, it should be noted that the increase in building construction area from 2010 to 2015 is also assumed to be equal to the population increase. The reason behind

such an assumption and the performance of the residential building stock model will be further evaluated in the Results and Discussion section.

After getting the population living in each urbanity of each province amplified to year 2015, now this urbanity-level based population data can be disaggregated into the geo-coded grids in 2015 GHS population density profile by using the apportionment weight (namely factor $F3$ in step 1-4 of Fig. 1). $F3$ is defined as the population share

of each grid relative to the summed population from grids within the same urbanity level of each province.



**2.3: Step 2-Disaggregate population and building related census data from urbanity level into grid level.**

As explained in Section 2.2, by multiplying the original building related records extracted from the 2010-census with factor *F0, F1, F2* and *F3* in Step 1 of Fig. 1, the population in each grid living in building types grouped by number of storey (1, 2-3, 4-6, 7-9, ≥10) or by construction material (steel/RC, mixed, other, brick/wood) can be
derived.

To estimate the residential building stock value, the number of buildings with combination of both storey class and construction material need to be derived. Initially, from the five storey classes (1, 2-3, 4-6, 7-9, ≥10) and the four building material classes (steel/RC, mixed, other, brick/wood), there will be 20 building sub-types. In the following description, we will first introduce how to reduce the principal number of building sub-types from 20 to
17 based on necessary assumption. Then we will estimate the number of population living in each of the 17 building sub-types. Based the information on average floor area per capita in each urbanity level (as given in the Short Table of the 2010-census), the total floor area of each of the 17 building sub-types in each grid can be derived. Finally, for each building sub-type, their replacement value emerges from a multiplication of the floor area with the construction price.

It is widely observed that most brick/wood buildings are with quite low height (1 or 2-3 storey), while steel/RC buildings are generally quite high with height of 10-storey or above. Therefore, it is further assumed that for "brick/wood" building type, there are only two storey classes (1, 2-3). While for "steel/RC", "mixed", and "other" building types defined in the 2010-census, all five storey classes (1, 2-3, 4-6, 7-9, ≥10) are available (namely *Assumption 1* in Step 2-1 of Fig. 1). Thus, the building sub-types in each grid are reduced from 20 to 17. The list
of these 17 building sub-types is given in Table 3.

Currently, we know from Step 1 for instance in each grid the number of population living in buildings of the five storey classes, but do not know for each storey class how the population are distributed in the classes of the four construction materials. Also, we know for instance how many people live in steel/RC buildings but do not know how they are distributed into the five storey classes. The derivation of the number of population in each of the 17
building sub-types requires to find 17 unknowns from 9 equations. In order to solve this underdetermined linear problem, further reasonable approximations need to be made (namely *Assumption 2* in Step 2-2 of Fig. 1) to make sure that in each grid the sum of population living in the 17 building sub-types is equal to the population living in building types grouped by construction material or by storey class.

From here, the population living in each of the 17 building sub-types is derived by a series of distribution steps
based on a prioritized ranking of building types and storey class from the aggregated inputs:

1.  In each grid, brick/wood buildings are first placed into 1 storey class and subtracted from the total amount of brick/wood buildings.

2.  Remaining brick/wood buildings are placed into 2-3 storey class.

3.  10 storey values are placed in steel/RC class as a start as they are assumed to not be "mixed" masonry
class.




4.  Similarly, the remaining steel/RC buildings are proportioned to other storey classes from highest to lowest, assuming that the least population in steel/RC would be in 1 storey class.

5.  For "other" buildings, they are distributed into each of the five storey classes, based on the proportions of remaining buildings in each storey class (all four construction materials are considered) and the ratio

between "other" buildings and "other + mixed" buildings.

6.  For "mixed" buildings, they are distributed each of the five storey classes, based on the proportions of remaining buildings in each storey class (all four construction materials are considered) and the ratio between "mixed" buildings and "other + mixed" buildings.

The MATLAB script illustrating the above multi-variate equation solving process is provided in Data/Code

Availability section.

**2.4: Step 3-Derive the number of people living in each of the 17 building sub-types**

With necessary assumption and approximation and by solving the multi-variate equations mentioned in Section 2.3, the population living in each of the 17 building sub-types can be derived for each grid. In the Short Table of 2010-census, the average residential floor area per capita is also given for each urbanity level of each province

(namely factor $F4$ in Step 3-1 of Fig. 1). Therefore, the floor area of the 17 building sub-types in each grid can be directly derived. Comparison between the modelled floor area with statistical yearbook recorded residential floor area for Shanghai will be performed in the Results and Discussion section.

With the building floor area known in each grid, to model the building stock value, another key component is the replacement value per square meter of each of the 17 buildings sub-types (namely factor $F5$ in Step 3-2 of Fig. 1).

Given the specialty/uniqueness of the building classification in this study, there is no official construction prices evaluated for the building types used here. Therefore, the unit construction price for each of the 17 building sub-types is derived (as listed in Table 3) by averaging the values given from different sources (e.g. 2015 China Construction Statistical Yearbook, the World Housing Encyclopedia, real-estate agency reports etc.). It should be noted that, due to the disparity of urbanization level, the actual construction price varies across urbanity levels and

provinces in mainland China. Therefore, when applying the residential building stock model to target area for risk analysis, the construction price should be modified accordingly. In this study, the set of averaged unit construction prices for the 17 building sub-types listed in Table 3 is used mainly to initially evaluate the replacement value of the residential building stock in each geo-coded grid.

**2.5: Step 4-Derive the replacement value of the 17 building sub-types in each grid.**

As elaborated in Step 3, after multiplying the floor area with unit construction price, the replacement value of the 17 building sub-types within each grid can be evaluated. By summing up the replacement value of all the geo-coded grids, the overall residential building stock value in mainland China can also be derived (in RMB of 2015 current prices). It is worth to emphasize that in this residential building stock model, the term "building replacement value" is used, which refers to the amount that will be needed to rebuild a property exactly as it was

prior to its destruction regardless of any depreciation due to its age, i.e. gross capital stock (Gunasekera et al., 2015).





## 3. Results and Discussion

### 3.1: Results----urbanity-level (urban/township/rural) based sum of modelled floor area and replacement value

Following the efforts of extensive data survey, collection and processing, with the modelling components and steps being explained in detail in Data Sources and Methodology section, a high-resolution (1km×1km) building stock model for mainland China targeted for future seismic risk assessment is established by disaggregating urbanity-level based census data into grid level. Since the census data are mainly related to residential buildings, the model developed is thus particularly for residential buildings. As listed in Table 4, the modelled residential building floor
area and replacement value (unit: RMB, in 2015 current prices) in each grid are aggregated into urbanity level (urban/township/rural) for each province.

In 2015, the total modelled residential building floor area for mainland China reaches 42.64 billion $m^2$. By applying the same replacement price for the same building sub-type (in total 17) in all the urban/township/rural areas of the 31 provinces, the initially modelled residential building stock value in whole mainland China is approximately to
265 be 77.6 trillion RMB (in 2015 current prices). It is clear that, like all other building stock, the Chinese building stock is a complicated economic, physical and social system (Yang and Kohler, 2008). The vacant building stock is also accounted for, thus is seen for places like New Ordos City. The economic disparity and geographic climatic diversity are widely spanned and the standardization in building construction also varies in different periods. Therefore, it is mainly for calculation convenience that this study applies the same unit construction price for all
270 the provinces and all the urbanity levels. However, to improve accuracy in future seismic risk assessment, the unit construction price of specific building types in the target study area should be adjusted accordingly.

### 3.2: Discussion

In this study, the building stock model is established through the disaggregation of urbanity-level based 2010-census data into grid level by using 2015 GHS population density profile as the proxy. Due to the approximation
and assumption made in this modelling process, the reasonability and consistency of the modelled results need to be cross validated. Due to the typical lack of official statistics on accumulated building stock value from the government (Wu et al., 2018), direct comparison of the modelled floor area and replacement value with that from census or statistical yearbooks for the whole mainland China is not available. Instead, the estimated stock value in previous studies is resorted to compare their modelled results with that in this study at provincial level.

### 3.2.1: Provincial-level based comparison between the modelled building value in this study and the net capital stock value estimated in Wu et al. (2014)

Previous studies on the capital stock estimation of mainland China mainly employed the perpetual inventory method (PIM), in which economy indicators e.g. gross fixed capital formation, total investment in fixed assets etc. were used. In general, these estimations are almost exclusively limited at national or provincial levels (Wu et al.,
2014). Such coarse spatial resolution forms a major obstacle in applying the model in disaster loss estimation, due to the mismatch between the hazard extent and the administrative boundary. To better address this gap, Wu et al., (2014) estimated the net capital stock value (WKS) for 344 prefectures in mainland China by using the perpetual



inventory method (PIM). In which, the WKS value (as listed in their Table A1) was calculated in 2012 current prices, with the depreciation of all exposed assets (i.e. residential and non-residential building structures, tools, machinery, equipment and infrastructure) being considered.

To better evaluate the reliability and consistency of the modelled results in this study, the estimated net capital stock value in Wu et al. (2014) for prefectures within the same province is aggregated into provincial level first, as shown in Table 4. The ratio between the modelled residential building stock value in this study (represented by "A") and the net capital stock value (represented by "C") in Wu et al. (2014) for each province is calculated in column "(A)/(C)" of Table 4 for straightforward comparison. The value of (A)/(C) varies within the range of **0.31-0.65**, which indicates the high consistency between the residential building replacement value modelled in this study in each province and the net capital stock value (for residential and non-residential buildings, infrastructure and other exposed elements) estimated in Wu et al. (2014), in spite of the differences in methodology and assumptions used in these two studies.

### 3.2.2: District-level based comparison between the modelled building floor area in this study and that recorded in statistical yearbook for Shanghai

A grid-level building stock model for Shanghai was developed in Wu et al. (2019), by disaggregating the district-level building floor area using building footprint map (extracted from high-resolution remote sensing data), combined with LandScan population density data as well as a financial appraisal of construction price according to building occupancy. However, Wu et al. (2019) did not separate residential floor area from non-residential floor area. Therefore, direct comparison of the modelled results from this study with their outputs is not available. On the other hand, yearbook records of the district-level residential and non-residential floor area, that were used in their study for model performance evaluation, turn out to be a good reference for this study to evaluate the modelled results at district-level, which can be extracted from Shanghai 2015 Statistical Yearbook.

To compare with the district-level residential floor area records in Shanghai statistical yearbook, the modelled floor area in each grid in Shanghai (Fig. 2) is aggregated into district level (as summarized in Table 5). As can be seen from Fig. 2 that grids with high floor area typically cluster in downtown area (including eight administrative districts, namely Yangpu, Hongkou, Zhabei, Putuo, Changning, Xuhui, Jing'an and Huangpu) and in Pudong district. This corresponds to the fact that these districts are the most developed in Shanghai. As can be further validated from the 3D-view of population distribution in panel (c) of Fig. 2, these districts also have the highest population density in Shanghai.

Table 5 gives a summary of the population in 2015 GHS population density profile, the modelled floor area (classified by storey classes) in this study, as well as the 2015 statistical yearbook recorded population and floor area for districts/counties in Shanghai. For more direct comparison, the initially modelled floor area (without adjustment) and the yearbook recorded floor area in each district of Shanghai are plotted in Fig. 3. The correlation between the initially modelled floor area and that recorded in yearbook turns out to be high, as indicated by the $R^2$ value (0.91). However, when it comes to the absolute floor area value, the total residential floor area modelled in Shanghai is around 808 km$^2$, while the yearbook recorded residential floor area is 611 km$^2$, which means the initially modelled results is overpredicted (around 1.3 multiples of the yearbook records). Therefore, additional





efforts are required to adjust the initially modelled results, to make the modelled floor area in each district more reasonably distributed and to de-amplify the overprediction of the overall modelled results.

As discussed in the modelling process in the Data Sources and Methodology section, it is clear that the disaggregation of urbanity level floor area into each grid has not integrated the development disparity of districts/counties within the same province. Therefore, the initially modelled floor area will be firstly rectified by

using the index of Uniform Construction Cost (UCC) to reflect the development inequality across districts in Shanghai, which has been used in previous studies (e.g. Gunasekera et al., 2015). The UCC index of each district in Shanghai is derived from the population and per capita GDP in 2015, which is defined as the triple root of the ratio between each district's GDP/capita and the average GDP/capita of Shanghai in 2015. As listed in Table 7, the higher the UCC index value, the more developed the corresponding district.

By multiplying the initially modelled floor area value with the UCC index in each district of Shanghai, the overall modelled floor area turns from 808 km$^2$ to 785 km$^2$. Although the overall floor area changes slightly, the application of UCC adjustment reallocates the floor area in each district, making it more consistent with the development level of each district. Meanwhile, compared with the recorded floor area of 611 km$^2$ for Shanghai in the yearbook, the UCC index adjusted floor area of 785 km$^2$ remains to be an obvious overprediction. Thus, de-

amplification adjustment is needed as well. By checking the whole modelling process in Fig. 1 carefully, it is found out that the overprediction of the modelled floor area for Shanghai may be attributed to the use of amplification factor ($F2$ in Step 1-3 of Fig. 1). $F2$ is used to synchronously amplify the building related census data from year 2010 to 2015 level. Mathematically it is equal to the ratio between 2015 GHS population and 2010-census population for each urbanity level of each province. For example, the amplification factor $F2$ in Shanghai is

1.33/1.34/1.29 for urban/township/rural level, respectively.

In reality, the increase of population in each urbanity level may not necessarily lead to the proportional increase of its residential floor area. Therefore, de-amplification of the initially modelled area for the whole Shanghai is attempted here. The derivation of the de-amplification factor of Shanghai is achieved by summarizing the product between the amplification factor of each urbanity level ($F2$) and its modelled floor area proportion. As shown in

Table 6, the final de-amplification factor of Shanghai is 1.32.

After further applicating the de-amplification factor to the modelled floor area in Shanghai (which is 785 km$^2$ in total with UCC index adjustment), the final modelled floor area in each district of Shanghai is listed in Table 5. To better illustrate the difference between the initially modelled floor area and that adjusted by UCC index and de-amplification factor in each district of Shanghai, the comparison of modelled floor area (before and after

adjustment) with statistical yearbook recorded floor area is plotted in Fig. 3. After adjusting the modelled floor area for each district of Shanghai with UCC index and de-amplification factor 1.32, for eight out of ten districts, the modelled floor area has a better match with yearbook records, except that Pudong district and Downtown are downwards deviating from the yearbook records after applying the de-amplification factor (Fig. 3). This is also easy to understand, since Pudong and Downtown are the most prosperous areas in Shanghai with increasing

population inflow. Therefore, the increase of residential floor area in these two districts can be regarded as proportional to the increase of population. Thus, the de-amplification adjustment may not be appropriate for these two districts.



However, in general, after the adjustment of initially modelled floor area by UCC index and the de-amplification factor, the overall modelled floor area in Shanghai turns to 594 km$^2$, only a 3% difference compared with the
statistical record of 610.9 km$^2$ (Table 5). As can be more clearly seen from Fig. 3, the value of the correlation indicator R$^2$ improves from 0.91 (before adjustment) to 0.94 (after adjustment). This further indicates the reasonability of the adjustment made and the reliability of the modelled residential floor area in this study for Shanghai.

### 3.2.3: Application of the model to seismic loss estimation

Since the model developed in this study is mainly targeted for seismic risk analysis, the performance of the model is further evaluated by its application to the estimation of empirical loss in scenario earthquake.

The hazard component used for this loss assessment test is the macro-seismic intensity map of the 2008 Wenchuan Ms8.0 earthquake (Fig. 4), which was issued by the China Earthquake Administration (CEA) based on the post-earthquake field investigations. The vulnerability function used was the empirical loss function developed in
Daniell (2014, Page 242) for mainland China. This empirical loss function was developed based on reported seismic damage and loss related to earthquakes that occurred in mainland China in the past few decades. Such information was retrieved through extensive collection of damage and loss records from journals, books, reports, conference proceedings and even newspapers, etc. Finally, based on the modelled residential building floor area in this study for Sichuan province and the unit construction price listed in Table 3, the estimated empirical loss to
residential buildings caused by the recurrence of the Wenchuan Ms8.0 earthquake is around 432 billion RMB (in 2015 current price). The distribution of loss ratio, i.e., the ratio between the estimated loss and the residential building stock value in counties/districts of Sichuan Province that were damaged in the Wenchuan Ms8.0 earthquake is shown in Fig. 5.

In other reports and studies on the loss assessment of Wenchuan earthquake, e.g. in Yuan (2008), the estimated
loss to residential buildings was around 170 billion RMB (in 2008 current price). The officially issued loss estimated by the Expert Panel of Earthquake Resistance and Disaster Relief (EPERDR, 2008) to residential buildings in Sichuan province was around 98.3-435.4 billion RMB, with the median around 212.32-247.25 billion RMB (in 2008 current price). It should be noted that in those studies, the unit construction price used for rural/urban/township buildings replacement was around 800-1500 RMB per square meter, which is 1/2.5-1/1.5 of
the unit construction price used in this study as listed in Table 3. To reduce the gap in construction price used in this study and in previous studies, the estimated loss value (432 billion RMB) in this study is further divided by 1.5-2.5, so that the final loss estimate is around 144-288 billion RMB (in 2015 current price). Therefore, the estimated loss range, based on the buildings stock model developed in this study and the empirical loss function developed in Daniell (2014), is quite compatible with that given in previous studies. This compatibility further
validates the robustness of our residential building stock model. Thus, the grid level building stock model developed in this study can be regarded as a reliable component input for further seismic risk assessment.





## 4. Conclusion

In this paper, a grid-level residential building stock model (in terms of floor area and replacement value) targeted for seismic risk analysis for mainland China is developed, by using 2015 GHS population density profile as the proxy and by disaggregating the urbanity level 2010-census data into 1km×1km scale for each province. To evaluate the model performance, the residential building stock value is compared with the net capital stock value estimated in Wu et al. (2014) using a perpetual inventory method at provincial level. The modelled stock value in these two studies are indeed quite consistent for all the 31 provinces in mainland China. Furthermore, district level comparison of the residential floor area developed in this study with records from the statistical yearbook of Shanghai is also conducted. It turns out that the floor area developed in this study is highly compatible with the floor area recorded in the yearbook of Shanghai. An adjustment to the modelled results is applied in order to more reasonably reflect the development disparities among districts within Shanghai. To further validate the performance of the model in seismic risk assessment, an empirical loss estimation for a recurrence of the 2008 Wenchuan M8.0 earthquake is performed. By reducing the gap in unit construction price used in this study and in previous studies, the overall estimated loss compares well with loss derived from damage reports based on field investigation. These results indicate the reliability of the geo-coded grid-level residential building stock model developed in this study. It is flexible for updates when more detailed census or statistics data are available, and it can be conveniently combined with hazard data and vulnerability information for risk assessment.

A limitation of this work is the focus on the residential building stock, as this exposure is accessible with the detailed census data. Although the damage to and the collapse of buildings is the main cause of fatalities and economic loss, damage to non-residential buildings (office, school, hospital, hotel, warehouse, factory, shop, cinema, etc.) as well as to life-line networks, infrastructures are not negligible. Therefore, future efforts should be made to estimate the stock value of non-residential buildings and infrastructures at risk. Furthermore, the replacement value developed in this study did not integrate the depreciation of the exposed buildings. Future work should target these deficiencies to better serve seismic risk analysis and loss mitigation strategies.

### Data/Code Availability

1. 2015 Global Human Settlement (GHS) population density profile: http://data.europa.eu/89h/jrc-GHS-ghs_pop_gpw4_globe_r2015a.

2. 2010 China Sixth Population Census Tabulation: http://www.stats.gov.cn/tjsj/pcsj/rkpc/6rp/indexch.htm

3. 2015 China Statistical Yearbook On Construction:
http://tongji.cnki.net/kns55/navi/YearBook.aspx?id=N2017020307&floor=1

4. 2015 Shanghai Statistics Yearbook: http://tjj.sh.gov.cn/html/sjfb/201701/1000201.html

5. Global Administrative Areas (GADM): www.gadm.org

6. An example illustrating the multi-variate equation solving process in Data Sources and Methodology section (the following two files are also available from the online supplementary document):



(a) Input file: https://www.jianguoyun.com/p/DdOYRvoQgPb4Bhi-hdUB

(b) MATLAB script: https://www.jianguoyun.com/p/DcAageEQgPb4BhjHhdUB

## Author contribution

DX conducted the data collection and preparation, results analysis, model validation and prepared the draft
manuscript. JD guided the data collection and preparation process, developed the modelling methodology and
performed the calculation and co-analysed the results. HT and FW supervised the project and provided advice and
feedback in the process. All authors contributed to the revision of the manuscript.

## Competing interests

The authors declare that they have no conflict of interests.

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

Scale: 0 2.5 5 7.5km

Assign urbanity attribute to the grids in China population density profile developed by Global Human Settlement (GHS) project in 2015, based on the population λ in each grid with 1 km×1 km resolution. The urbanity level is assigned according to the following criteria:

rural: λ < 2736
township: 2736 ≤ λ < 4927
urban: λ ≥ 4927

**step 1**

2010-census recorded number of families living in following building types

grouped by building storey
grouped by construction material

From 2010-census, extract number of families living in building types grouped by building storey (5 classes) or by construction material (4 classes), within each urbanity level of each province.

| 1 | 2-3 | 4-6 | 7-9 | ≥10 |

brick/wood | steel/RC | mixed | other

1-1 Amplify family number in 2010-census data from 10% to 100% population } F0 = 10 (amplification factor)
1-2 Multiply average number of person per family) (urban, township, rural) } F1 (2010-census)
1-3 Amplify from 2010-census derived population (urban, township, rural) } F2 (amplification factor)
1-4 2015 GHS population share of each grid relative to the summed population from grids with the same urbanity in each province } F3 (apportionment weight)

**step 2**

The number of population living in each building type classified by storey (5 classes) or construction material (4 classes) is disaggregated into geo-coded grids.

2-1 Assume "brick/wood" only has storey classes of 1 and 2-3 and "steel/RC, mixed, other" have five storey classes. In total, 17 sub-types of building can be classified. } Assumption 1

2-2 Assume ≥10-storey buildings are mainly RC buildings, then solve multi-variate equation to derive the number of population living in each of the 17 building sub-types. } Assumption 2

**step 3**

Derive the number of population living in each of the 17 building sub-types in each grid.

| | | | |
|---|---|---|---|
| BRIWOMC1 | 4963 | MIXEDMC23 | 7615.89 |
| BRIWOMC23 | 364 | MIXEDMC46 | 0 |
| STLRCMC1 | 0 | MIXEDMC49 | 0 |
| STLRCMC23 | 808.78 | MIXEDMC10 | 0 |
| STLRCMC46 | 555.68 | OTHERMC1 | 0 |
| STLRCMC79 | 7.69 | OTHERMC23 | 208.00 |
| STLRCMC10 | 41.41 | OTHERMC46 | 0 |
| MIXEDMC1 | 0 | OTHERMC79 | 0 |
| | | OTHERMC10 | 0 |

3-1 Multiply floor area per capita (urban, township, rural) } F4 (2010-census)
3-2 Multiply replacement price per square meter for each of the 17 building sub-types } F5 (averaged price)

**step 4**

Derive the replacement value of the 17 building sub-types in each grid.

| | | | |
|---|---|---|---|
| VBRIWOMC1S | 334511797.35 | VMIXEDMC23 | 751079028.88 |
| VBRIWOMC23 | 28101844 | VMIXEDMC46 | 0 |
| VSTLRCMC1S | 0 | VMIXEDMC49 | 0 |
| VSTLRCMC23 | 103691633.84 | VMIXEDMC10 | 0 |
| VSTLRCMC46 | 73300593.35 | VOTHERMC1 | 0 |
| VSTLRCMC79 | 1086502.52 | VOTHERMC23 | 19145546.05 |
| VSTLRCMC10 | 6126079.70 | VOTHERMC46 | 0 |
| VMIXEDMC1S | 0 | VOTHERMC79 | 0 |
| | | VOTHERMC10 | 0 |

4-1 Note: the same replacement price per square meter is used for the same building sub-type in urban/township/rural grids without further adjustment.

**Figure 1:** Flowchart of the residential building stock modelling process for mainland China. Detailed explanations of each component and step are given in the Data Sources and Methodology section.

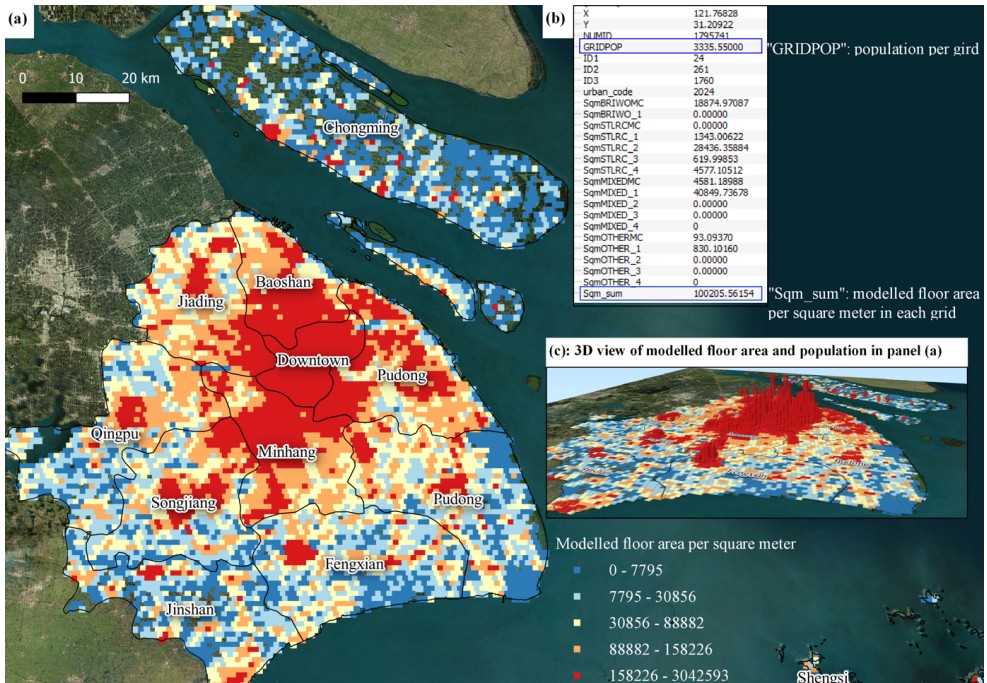

**Figure 2**. An example illustrating the modelled floor area for Shanghai: (a) the distribution of modelled floor area in each grid with resolution of 1km×1km; (b) This table shows the modelled floor area (unit: m$^2$) of the 17 building sub-types in one example grid, as well as the total population "GRIDPOP" and the total modelled floor area "Sqm_sum" in each grid; (c) the 3D view of the modelled floor area and the population distribution (the height of cuboid in each grid is proportional to its population density; the colour of each cuboid represents the modelled floor range). This figure is plotted using QGIS platform (https://qgis.org/en/site/) and the background aerial map is provided by Bing map service (Copyright: under the © Microsoft® BingTM Maps Platform APIs' terms of use, last updated May 2018).

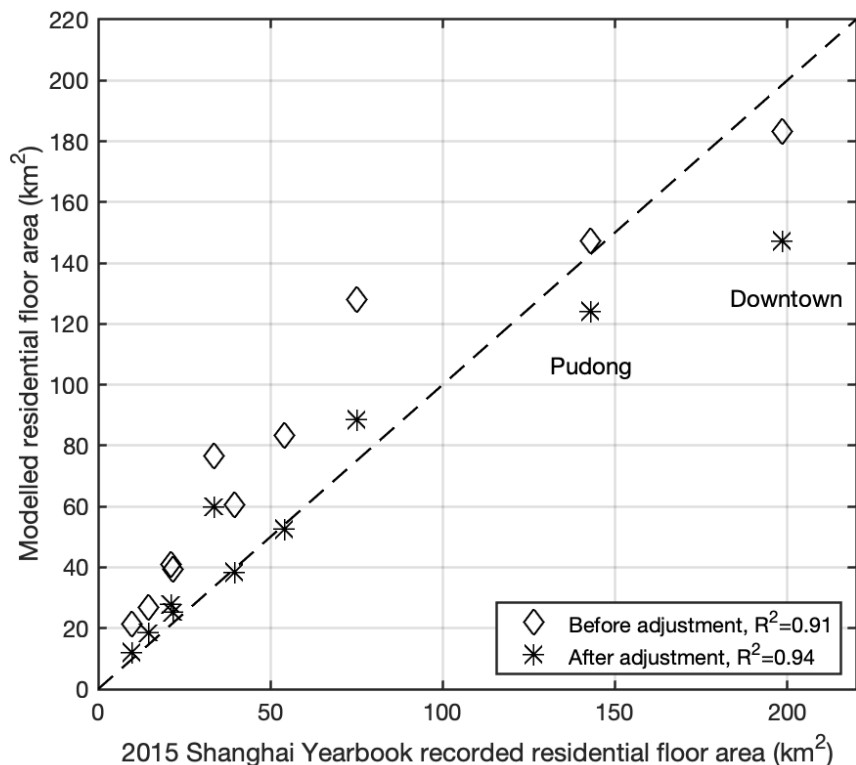

**Figure 3**: Comparison of the modelled floor area (before and after adjustment) with 2015 Shanghai Yearbook recorded floor area for each district of Shanghai.
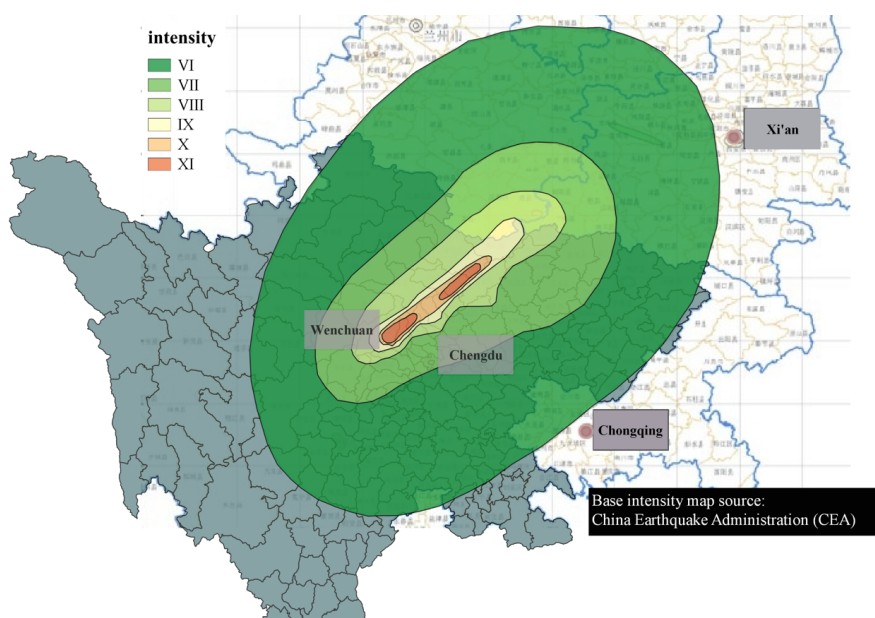

**Figure 4**. Macro-seismic intensity map of 2008 Wenchuan Ms8.0 earthquake, modified after the original map issued by China Earthquake Administration (CEA).

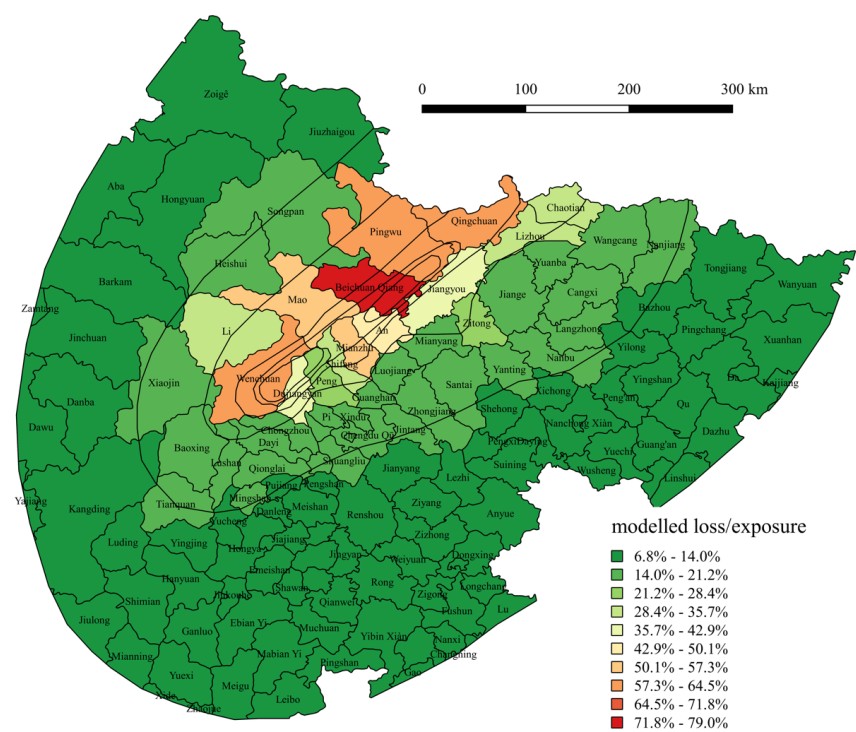

**Figure 5**. Distribution of estimated residential building loss ratio (the ratio between loss and exposed stock value) in affected districts/counties in Sichuan Province, by using the exact intensity map of the 2008 Wenchuan Ms8.0 earthquake as the hazard input. The black contours indicate the intensity levels within Sichuan Province (as shown in Figure 4).


**Table 1**: Main data sources used in this study. Accesses to these data are provided in the Data/Code Availability section.

| Data source | Data description | Resolution | Data location | Indicator in the text | Notes |
|---|---|---|---|---|---|
| 2010 China Sixth Census Short Table | Overall population | urban/township/rural level for each of the 31 provinces/municipalities in mainland China; (the urbanity level in the census is defined according to the administrative belonging of the surveyed population) | Table 1-1a, 1-1b, 1-1c | N/A | Based on surveys of 100% of the population in mainland China |
| 2010 China Sixth Census Long Table | Number of families living in buildings grouped by usage (residential, commercial, mixed) Number of families dwelled in buildings grouped by storey number (1, 2-3, 4-6, 7-9, ≥10) Number of families dwelled in buildings grouped by construction material (steel/RC, mixed, other, brick/wood) | | Table 9-1a, 9-1b, 9-1c | N/A | Based on surveys of 10% of the overall population in mainland China |
| 2010 China Sixth Census Short Table | Average population per family | | Table 1-1a, 1-1b, 1-1c | D3 of Fig. 1 | Based on surveys of 100% of the population in mainland China |
| | Average residential floor area per person (unit: square meter) | | Table 1-14a, 1-14b, 1-14c | D4 of Fig. 1 | |
| 2015 GHS population density profile | provides the population density in each geo-coded grid | 1km×1km | N/A | λ | The original resolution is 250m×250m and was resampled to 1km×1km |
| 2015 Shanghai Statistics Yearbook | GDP and population in each district | District level | Page 495-545 | N/A | To derive the uniform construction cost (UCC) |
| 2015 China Construction Yearbook | Yearly construction value added in each province | Provincial level | Table 1-2 | N/A | These data are used to evaluate modelled building stock value |



Natural Hazards and Earth System Sciences

EGU

Discussions



**Table 2:** Residential building stock modelling related data extracted from the Tabulation of the 2010 Population Census of the People's Republic of China (abbreviated as "2010-census").

| "Urbanity" +"0"+ Prov_ID | Province | 2015 GHS Pop. | Floor area per capita | Person per family | Number of families grouped by occupancy | | | Number of families grouped by building height (storey class) | | | | | Number of families grouped by construction material | | | | Amplification Factor (F2 in Fig.1, to amplify 2010-census data to 2015 level) |
|---|---|---|---|---|---|---|---|---|---|---|---|---|---|---|---|---|---|
| | | | | | living | commercial | mixed | 1 | 2-3 | 4-6 | 7-9 | ≥10 | steel/RC | mixed masonry | brick-wood | others | |
| | | | | | | | | **urban** | | | | | | | | | |
| 1001 | Anhui | 12162978 | 29.42 | 2.71 | 331730 | 9035 | 287 | 44093 | 82489 | 175486 | 20922 | 17775 | 135377 | 176462 | 26705 | 2221 | 1.32 |
| 1002 | Beijing | 18597540 | 27.81 | 2.40 | 517975 | 6482 | 988 | 127740 | 33290 | 193270 | 21919 | 148238 | 226367 | 212873 | 83192 | 2025 | 1.47 |
| 1003 | Chongqing | 8391462 | 29.77 | 2.65 | 258417 | 3956 | 247 | 17185 | 39448 | 85383 | 81270 | 39087 | 131656 | 112494 | 13433 | 4790 | 1.21 |
| 1004 | Fujian | 12699884 | 30.29 | 2.70 | 360721 | 13488 | 736 | 30557 | 97680 | 135725 | 30332 | 79915 | 213350 | 124702 | 23948 | 12209 | 1.25 |
| 1005 | Gansu | 5282457 | 26.69 | 2.68 | 160717 | 3134 | 107 | 24489 | 21076 | 75051 | 9074 | 78731 | 66665 | 15057 | 3398 | 1.20 |
| 1006 | Guangdong | 56519993 | 26.37 | 2.63 | 1466895 | 34218 | 513 | 152601 | 299326 | 453172 | 412115 | 183699 | 663772 | 748196 | 76682 | 12465 | 1.43 |
| 1007 | Guangxi | 8478357 | 30.71 | 2.93 | 238044 | 5912 | 264 | 26305 | 53876 | 99335 | 52485 | 11955 | 86601 | 138730 | 16271 | 2354 | 1.18 |
| 1008 | Guizhou | 5485811 | 25.94 | 2.82 | 157713 | 5141 | 19 | 17373 | 38055 | 50766 | 49256 | 11955 | 52485 | 78055 | 1262 | | 1.19 |
| 1009 | Hainan | 2327452 | 25.42 | 3.17 | 56383 | 1602 | 68 | 9674 | 14288 | 13787 | 13124 | 7112 | 41510 | 108814 | 4948 | 713 | 1.26 |
| 1010 | Hebei | 14836541 | 30.10 | 2.95 | 419978 | 3950 | 96 | 100741 | 42944 | 230919 | 29889 | 19435 | 155581 | 211716 | 54745 | 1886 | 1.19 |
| 1011 | Heilongjiang | 14367419 | 23.72 | 2.58 | 455996 | 6911 | 418 | 122051 | 20020 | 130862 | 173283 | 16691 | 163427 | 188650 | 104208 | 6622 | 1.20 |
| 1012 | Henan | 18527056 | 34.02 | 3.05 | 521036 | 7612 | 215 | 79535 | 122569 | 244091 | 64920 | 17533 | 190648 | 307902 | 28268 | 1830 | 1.15 |
| 1013 | Hubei | 17537483 | 33.22 | 2.82 | 502439 | 12733 | 349 | 40937 | 132838 | 179474 | 126270 | 35653 | 180316 | 298109 | 33900 | 2847 | 1.21 |
| 1014 | Hunan | 12911981 | 33.45 | 2.89 | 358447 | 9813 | 501 | 32935 | 92165 | 160007 | 62887 | 20266 | 132713 | 201615 | 31404 | 2528 | 1.21 |
| 1015 | Jiangsu | 30857658 | 33.86 | 2.81 | 876264 | 14961 | 802 | 129293 | 224580 | 412115 | 65052 | 60185 | 325288 | 469388 | 92721 | 3828 | 1.23 |
| 1016 | Jiangxi | 7844695 | 29.76 | 3.19 | 201690 | 3594 | 201 | 17052 | 46727 | 85663 | 48457 | 7385 | 111658 | 76679 | 15396 | 1551 | 1.20 |
| 1017 | Jilin | 10270924 | 25.21 | 2.62 | 329782 | 4910 | 1777 | 59861 | 13029 | 149906 | 96067 | 15829 | 157788 | 108325 | 48852 | 1727 | 1.17 |
| 1018 | Liaoning | 22172958 | 25.76 | 2.57 | 768884 | 7122 | 843 | 111439 | 366106 | 211530 | 58885 | 321935 | 381031 | 71386 | 1654 | | 1.11 |
| 1019 | Inner Mongolia | 8302698 | 24.86 | 2.67 | 251738 | 6951 | 631 | 84432 | 24977 | 133932 | 11690 | 3658 | 105902 | 87092 | 61924 | 3771 | 1.20 |
| 1020 | Ningxia | 2215109 | 28.38 | 2.71 | 64336 | 1829 | 29 | 10922 | 7958 | 44770 | 1313 | 1202 | 24606 | 34483 | 6352 | 724 | 1.23 |
| 1021 | Qinghai | 1470242 | 27.77 | 2.74 | 41342 | 1229 | 62 | 4877 | 8035 | 20737 | 6292 | 2630 | 13527 | 26113 | 2415 | 516 | 1.26 |
| 1022 | Shaanxi | 9021036 | 28.81 | 2.70 | 269044 | 4820 | 362 | 33723 | 56478 | 122687 | 37356 | 23620 | 89287 | 173753 | 8694 | 2130 | 1.22 |
| 1023 | Shandong | 28921044 | 32.41 | 2.80 | 855282 | 15616 | 242 | 252471 | 88326 | 432226 | 67205 | 30670 | 348873 | 356038 | 161295 | 4692 | 1.19 |
| 1024 | Shanghai | 20557127 | 25.11 | 2.52 | 604654 | 9991 | 928 | 60506 | 116799 | 304794 | 104766 | 104766 | 268377 | 249438 | 93734 | 3096 | 1.33 |
| 1025 | Shanxi | 9837996 | 25.77 | 2.88 | 282847 | 4319 | 87 | 53815 | 47879 | 157087 | 18683 | 9702 | 90187 | 163209 | 29124 | 4646 | 1.19 |
| 1026 | Sichuan | 15732199 | 30.70 | 2.67 | 499024 | 9628 | 630 | 47158 | 79975 | 198299 | 136824 | 46396 | 218827 | 247875 | 34088 | 7862 | 1.16 |
| 1027 | Tianjin | 10012251 | 25.51 | 2.65 | 237060 | 2606 | 167 | 34902 | 12083 | 143755 | 28570 | 20356 | 58333 | 156521 | 23467 | 1345 | 1.58 |
| 1028 | Xinjiang | 6578245 | 28.00 | 2.56 | 201621 | 2686 | 84 | 32261 | 24343 | 129144 | 12124 | 6435 | 88699 | 94628 | 18420 | 2560 | 1.26 |
| 1029 | Tibet | 289534 | 31.81 | 2.45 | 8394 | 973 | 7 | 2930 | 4798 | 1580 | 47 | 12 | 5449 | 2227 | 1020 | 671 | 1.26 |
| 1030 | Yunnan | 6531449 | 31.27 | 2.59 | 200602 | 7122 | 172 | 21262 | 45555 | 93027 | 36704 | 11176 | 102015 | 85386 | 13317 | 7006 | 1.21 |
| 1031 | Zhejiang | 21732071 | 30.97 | 2.54 | 675858 | 19305 | 774 | 80859 | 193447 | 332899 | 50666 | 37292 | 220048 | 393843 | 74559 | 6713 | 1.23 |
| | | | | | | | | **township** | | | | | | | | | |




| Code | Province | | | | | | | | | | | | | | | | |
|---|---|---|---|---|---|---|---|---|---|---|---|---|---|---|---|---|---|
| 2001 | Anhui | 13372970 | 32.20 | 2.95 | 355306 | 19130 | 477 | 144219 | 160370 | 67744 | 1426 | 677 | 95625 | 182264 | 91921 | 4626 | 1.21 |
| 2002 | Beijing | 1548063 | 33.20 | 2.52 | 41959 | 1129 | 143 | 21808 | 2812 | 16414 | 710 | 1344 | 20550 | 15964 | | 350 | 1.42 |
| 2003 | Chongqing | 6393138 | 34.91 | 2.73 | 187287 | 7816 | 357 | 35957 | 71385 | 40448 | 6157 | 6224 | 46425 | 112018 | 23805 | 12855 | 1.20 |
| 2004 | Fujian | 8616342 | 37.67 | 3.09 | 224647 | 11851 | 318 | 44154 | 105240 | 65529 | 2753 | 229 | 83984 | 28551 | 23313 | | 1.18 |
| 2005 | Gansu | 3949838 | 25.92 | 3.17 | 101071 | 5160 | 124 | 58128 | 13450 | 30226 | 4198 | 3722 | 31721 | 30839 | 34944 | 8727 | 1.17 |
| 2006 | Guangdong | 17954335 | 26.41 | 3.52 | 357650 | 15136 | 348 | 119634 | 160743 | 60743 | 27235 | 124661 | 175520 | 63890 | 8715 | | 1.37 |
| 2007 | Guangxi | 10216390 | 34.43 | 3.34 | 264485 | 12263 | 480 | 94666 | 111560 | 58971 | 549 | 42500 | 175149 | 42500 | 5370 | | 1.10 |
| 2008 | Guizhou | 6142030 | 28.39 | 3.12 | 159970 | 12522 | 41 | 65929 | 60006 | 34332 | 440 | 11785 | 89287 | 28725 | 10464 | | 1.14 |
| 2009 | Hainan | 1986929 | 23.78 | 3.42 | 45035 | 2592 | 51 | 26889 | 4359 | 15458 | 607 | 314 | 12356 | 14449 | 910 | | 1.22 |
| 2010 | Hebei | 17723090 | 30.74 | 3.40 | 454034 | 12232 | 203 | 338450 | 45232 | 73026 | 6074 | 90952 | 165751 | 204531 | 5032 | | 1.12 |
| 2011 | Heilongjiang | 7326077 | 22.67 | 2.63 | 230438 | 7764 | 526 | 152211 | 54825 | 54825 | 604 | 26869 | 70838 | 130084 | 10411 | | 1.17 |
| 2012 | Henan | 18079108 | 32.04 | 3.60 | 435993 | 14307 | 304 | 242151 | 151413 | 53669 | 391 | 2676 | 240373 | 114219 | 4012 | | 1.11 |
| 2013 | Hubei | 10287748 | 38.10 | 3.12 | 267951 | 11284 | 318 | 65151 | 136106 | 59020 | 2676 | 18152 | 75159 | 47125 | 6000 | | 1.18 |
| 2014 | Hunan | 15928705 | 36.74 | 3.18 | 413160 | 16084 | 1397 | 107304 | 216464 | 60305 | 2245 | 12926 | 103618 | 225168 | 92116 | 8342 | 1.16 |
| 2015 | Jiangsu | 17599234 | 39.53 | 3.00 | 493818 | 16021 | 436 | 194665 | 224247 | 86379 | 2249 | 99148 | 264939 | 142526 | 3226 | | 1.15 |
| 2016 | Jiangxi | 12539807 | 33.57 | 3.54 | 283781 | 10796 | 1125 | 57795 | 138466 | 80093 | 17102 | 1121 | 144491 | 98662 | 45425 | 5999 | 1.20 |
| 2017 | Jilin | 4483838 | 22.51 | 2.70 | 139477 | 4710 | 1966 | 90313 | 10161 | 37025 | 228 | 34567 | 73754 | 5399 | | | 1.14 |
| 2018 | Liaoning | 5202389 | 26.23 | 2.75 | 169863 | 5618 | 94 | 100064 | 11565 | 51923 | 9229 | 1500 | 51280 | 52098 | 69815 | 1088 | 1.08 |
| 2019 | Inner Mongolia | 5916056 | 24.38 | 2.74 | 172725 | 9637 | 1622 | 124351 | 14566 | 41832 | 191 | 1422 | 43195 | 35332 | 90983 | 12852 | 1.17 |
| 2020 | Ningxia | 1035570 | 24.82 | 3.14 | 25273 | 1397 | 58 | 16542 | 2590 | 7308 | 54 | 176 | 6140 | 12255 | 1166 | | 1.23 |
| 2021 | Qinghai | 1234007 | 21.94 | 3.06 | 28364 | 1806 | 1694 | 15491 | 4641 | 9622 | 386 | 30 | 8482 | 8928 | 2946 | 1166 | 1.27 |
| 2022 | Shaanxi | 8393227 | 28.85 | 3.05 | 218969 | 10349 | 295 | 103810 | 63776 | 53427 | 2172 | 6133 | 61288 | 115983 | 30075 | 21972 | 1.20 |
| 2023 | Shandong | 19633228 | 32.14 | 3.03 | 555539 | 16773 | 117 | 412345 | 53861 | 102936 | 2235 | 935 | 105549 | 177664 | 274908 | 14191 | 1.13 |
| 2024 | Shanghai | 3396024 | 30.25 | 2.45 | 100049 | 3066 | 715 | 24233 | 44272 | 29262 | 638 | 4710 | 35992 | 46750 | 19423 | 950 | 1.34 |
| 2025 | Shanxi | 8095334 | 25.43 | 3.24 | 208837 | 7124 | 292 | 128133 | 41454 | 42626 | 2929 | 819 | 49930 | 87194 | 66418 | 12419 | 1.16 |
| 2026 | Sichuan | 16239393 | 34.47 | 2.80 | 494678 | 24545 | 2048 | 133695 | 141458 | 9146 | 64579 | 144800 | 259633 | 80423 | 34367 | 1.11 | |
| 2027 | Tianjin | 1604748 | 29.64 | 2.98 | 36626 | 688 | 6 | 20978 | 1965 | 12727 | 559 | 1085 | 5896 | 13066 | 18217 | 135 | 1.44 |
| 2028 | Xinjiang | 3536191 | 26.04 | 2.75 | 95090 | 2368 | 50 | 57285 | 24233 | 32598 | 301 | 187 | 31109 | 21827 | 34576 | 9946 | 1.32 |
| 2029 | Tibet | 434071 | 33.52 | 2.89 | 10835 | 1334 | 69 | 5712 | 5333 | 1058 | 39 | 27 | 5633 | 2406 | 2961 | 1169 | 1.23 |
| 2030 | Yunnan | 9948973 | 30.04 | 3.29 | 249892 | 15089 | 538 | 95990 | 49076 | 49076 | 5598 | 540 | 85728 | 73181 | 58444 | 47628 | 1.14 |
| 2031 | Zhejiang | 14032915 | 38.53 | 2.66 | 435571 | 17019 | 321 | 78393 | 215994 | 143891 | 9590 | 4722 | 88524 | 262572 | 92204 | 9290 | 1.16 |

rural

| Code | Province | | | | | | | | | | | | | | | | |
|---|---|---|---|---|---|---|---|---|---|---|---|---|---|---|---|---|---|
| 3001 | Anhui | 33868749 | 34.04 | 3.12 | 972114 | 12697 | 1032 | 594442 | 384935 | 5062 | 259 | 113 | 122416 | 440296 | 399437 | 22662 | 1.10 |
| 3002 | Beijing | 3290554 | 35.39 | 2.76 | 85494 | 2139 | 89 | 81788 | 2698 | 2877 | 93 | 177 | 19546 | 63298 | 85298 | 1798 | 1.36 |
| 3003 | Chongqing | 13097499 | 42.04 | 2.72 | 436237 | 8496 | 810 | 215548 | 219389 | 6337 | 3076 | 383 | 34275 | 160849 | 146892 | 102717 | 1.08 |
| 3004 | Fujian | 16023424 | 41.24 | 3.16 | 447940 | 13851 | 615 | 152099 | 279096 | 27946 | 1860 | 190 | 105558 | 152003 | 108638 | 95592 | 1.10 |
| 3005 | Gansu | 16457361 | 21.94 | 3.89 | 444734 | 2789 | 233 | 434394 | 12043 | 911 | 94 | 81 | 23583 | 50990 | 233241 | 139709 | 0.94 |
| 3006 | Guangdong | 38073367 | 25.99 | 3.74 | 825588 | 7932 | 862 | 473821 | 328499 | 27016 | 3542 | 642 | 168179 | 388958 | 244088 | 32295 | 1.22 |
| 3007 | Guangxi | 28020960 | 34.82 | 3.47 | 788492 | 7837 | 834 | 494076 | 294396 | 7474 | 300 | 83 | 100152 | 424443 | 210891 | 60843 | 1.01 |
| 3008 | Guizhou | 22795976 | 27.92 | 3.29 | 657275 | 13176 | 244 | 526145 | 137494 | 5485 | 1206 | 121 | 80232 | 208026 | 247780 | 134413 | 1.03 |
| 3009 | Hainan | 4368909 | 21.29 | 3.63 | 109378 | 771 | 69 | 101212 | 8248 | 437 | 217 | 35 | 22309 | 16584 | 68949 | 2307 | 1.09 |
| 3010 | Hebei | 41534503 | 30.09 | 3.50 | 1138877 | 6755 | 525 | 1108487 | 32754 | 3591 | 510 | 290 | 65563 | 351042 | 689663 | 39364 | 1.04 |
| 3011 | Heilongjiang | 17284909 | 20.92 | 3.19 | 472849 | 3926 | 1647 | 469755 | 3174 | 2668 | 1148 | 30 | 5933 | 44163 | 339849 | 86830 | 1.13 |
| 3012 | Henan | 58426898 | 32.23 | 3.58 | 1593259 | 18790 | 715 | 1266614 | 341472 | 6231 | 554 | 178 | 170146 | 778487 | 632719 | 30697 | 1.01 |
| 3013 | Hubei | 28165214 | 38.64 | 3.40 | 805308 | 11381 | 807 | 395220 | 405959 | 12191 | 2267 | 1052 | 87280 | 373421 | 286599 | 69389 | 1.01 |
| 3014 | Hunan | 37755133 | 34.27 | 3.54 | 1008324 | 9900 | 2170 | 496152 | 516168 | 5569 | 262 | 73 | 113888 | 408562 | 427367 | 68407 | 1.05 |



| | | | | | | | | | | | | | | | |
|---|---|---|---|---|---|---|---|---|---|---|---|---|---|---|---|
| 3015 | Jiangsu | 32006376 | 42.35 | 3.03 | 978352 | 13096 | 999 | 526012 | 17344 | 893 | 2817 | 77218 | 494838 | 411206 | 8186 | 1.06 |
| 3016 | Jiangxi | 26204945 | 33.81 | 3.86 | 627420 | 6578 | 1410 | 251425 | 8390 | 355 | 118 | 184327 | 209487 | 198186 | 41998 | 1.07 |
| 3017 | Jilin | 12897767 | 20.98 | 3.35 | 353543 | 2220 | 2523 | 347297 | 4561 | 676 | 59 | 11283 | 35524 | 274007 | 34949 | 1.07 |
| 3018 | Liaoning | 16672483 | 25.95 | 3.12 | 519784 | 3994 | 237 | 512930 | 6643 | 390 | 106 | 31856 | 123657 | 360371 | 7894 | 1.02 |
| 3019 | Inner Mongolia | 11385344 | 22.17 | 2.97 | 337168 | 4773 | 1167 | 331674 | 6301 | 77 | 245 | 10616 | 34647 | 206674 | 99004 | 1.12 |
| 3020 | Ningxia | 3527454 | 22.12 | 3.54 | 86461 | 1371 | 35 | 80927 | 1965 | 64 | 13 | 4944 | 9056 | 60381 | 13451 | 1.13 |
| 3021 | Qinghai | 3342860 | 18.51 | 4.06 | 71842 | 604 | 1521 | 69459 | 2789 | 7 | 10 | 2675 | 9718 | 36221 | 23832 | 1.11 |
| 3022 | Shaanxi | 20689727 | 31.22 | 3.54 | 572916 | 6711 | 497 | 481090 | 3360 | 348 | 230 | 60338 | 235474 | 142395 | 141420 | 1.01 |
| 3023 | Shandong | 49116344 | 31.95 | 3.07 | 1549890 | 8748 | 182 | 1511164 | 6807 | 399 | 103 | 77610 | 400711 | 1025247 | 55070 | 1.03 |
| 3024 | Shanghai | 2871449 | 38.83 | 2.37 | 9072 | 1752 | 1153 | 31644 | 57352 | 49 | 264 | 8884 | 48551 | 33963 | 1326 | 1.29 |
| 3025 | Shanxi | 19386995 | 25.09 | 3.44 | 521669 | 4921 | 593 | 481296 | 6348 | 290 | 103 | 34053 | 138101 | 243316 | 111120 | 1.07 |
| 3026 | Sichuan | 47518958 | 36.63 | 3.10 | 1625052 | 36122 | 3253 | 1067677 | 574735 | 16573 | 764 | 147168 | 513785 | 611594 | 388627 | 0.92 |
| 3027 | Tianjin | 3007476 | 25.95 | 3.21 | 78318 | 570 | 30 | 74498 | 686 | 110 | 249 | 2325 | 7772 | 68306 | 485 | 1.19 |
| 3028 | Xinjiang | 13521011 | 22.35 | 3.55 | 314397 | 2226 | 115 | 309505 | 2663 | 82 | 28 | 11730 | 36704 | 207565 | 60624 | 1.20 |
| 3029 | Tibet | 2468309 | 27.55 | 4.95 | 44816 | 1260 | 718 | 27819 | 17858 | 360 | 26 | 2594 | 5152 | 23631 | 14699 | 1.07 |
| 3030 | Yunnan | 30987983 | 25.61 | 3.89 | 756974 | 10742 | 1276 | 461191 | 296513 | 6950 | 592 | 68863 | 112129 | 239753 | 346971 | 1.04 |
| 3031 | Zhejiang | 22254831 | 49.12 | 2.67 | 740469 | 17587 | 807 | 152558 | 544733 | 58732 | 1649 | 384 | 60829 | 419761 | 236627 | 40839 | 1.10 |



**Table 3**: Averaged construction price per square meter for each of the 17 building sub-types used in this study to estimate the building stock value in mainland China.

| Construction material | Storey class | Building type abbreviation | Construction price (RMB/m$^2$ in 2015 current price) |
|---|---|---|---|
| brick/wood | 1 | BRIWOMC1 | 2050 |
| | 2-3 | BRIWOMC23 | 2350 |
| steel/RC | 1 | STLRCMC1 | 3700 |
| | 2-3 | STLRCMC23 | 3900 |
| | 4-6 | STLRCMC46 | 4100 |
| | 7-9 | STLRCMC79 | 4300 |
| | ≥10 | STLRCMC10 | 4500 |
| mixed | 1 | MIXEDMC1 | 2800 |
| | 2-3 | MIXEDMC23 | 3000 |
| | 4-6 | MIXEDMC46 | 3200 |
| | 7-9 | MIXEDMC79 | 3400 |
| | ≥10 | MIXEDMC10 | 3600 |
| others | 1 | OTHERMC1 | 2600 |
| | 2-3 | OTHERMC23 | 2800 |
| | 4-6 | OTHERMC46 | 3000 |
| | 7-9 | OTHERMC79 | 3200 |
| | ≥10 | OTHERMC10 | 3400 |




**Table 4**: Modelled residential building floor area and replacement value for urban/township/rural area of 31 provinces/municipalities in mainland China and comparison with net capital stock value estimated in Wu et al. (2014) using perpetual inventory method (in this table, scientific notation is used to represent the large numbers).

| Prov_ID | Province | Initially modelled residential floor area (m²) urban | township | rural | (A): Initially modelled residential building stock replacement value (RMB in 2015 current price) urban | township | rural | (C): Net capital stock value modelled in Wu et al. (2014, RMB in 2012 current price) | (A)/(C) |
|---|---|---|---|---|---|---|---|---|---|
| 01 | Anhui | 3.57E+08 | 4.30E+08 | 1.15E+09 | 5.08E+11 | 4.97E+11 | 1.08E+12 | 3.86E+12 | 0.54 |
| 02 | Beijing | 5.16E+08 | 5.13E+07 | 1.16E+08 | 1.92E+12 | 1.48E+11 | 2.22E+11 | 3.85E+12 | 0.59 |
| 03 | Chongqing | 2.50E+08 | 2.23E+08 | 5.50E+08 | 5.63E+11 | 4.29E+11 | 8.25E+11 | 2.98E+12 | 0.61 |
| 04 | Fujian | 1.40E+08 | 2.46E+08 | 1.07E+09 | 3.61E+11 | 5.14E+11 | 2.02E+12 | 4.73E+12 | 0.61 |
| 05 | Gansu | 1.41E+08 | 1.02E+08 | 3.61E+08 | 2.31E+11 | 2.71E+11 | 1.14E+11 | 1.56E+12 | 0.39 |
| 06 | Guangdong | 1.11E+09 | 4.16E+08 | 1.40E+09 | 2.97E+12 | 8.05E+11 | 1.74E+12 | 1.07E+13 | 0.52 |
| 07 | Guangxi | 2.27E+08 | 2.94E+08 | 8.84E+08 | 8.05E+11 | 1.29E+12 | 3.24E+11 | 4.74E+12 | 0.51 |
| 08 | Guizhou | 1.42E+08 | 1.74E+08 | 6.36E+08 | 1.98E+11 | 2.19E+11 | 4.88E+11 | 2.08E+12 | 0.44 |
| 09 | Hainan | 1.82E+07 | 2.37E+07 | 1.43E+08 | 3.98E+10 | 3.87E+10 | 1.63E+11 | 7.86E+11 | 0.31 |
| 10 | Hebei | 3.90E+08 | 5.16E+08 | 1.33E+09 | 7.75E+11 | 8.23E+11 | 1.56E+12 | 6.82E+12 | 0.46 |
| 11 | Heilongjiang | 3.37E+08 | 1.65E+08 | 3.64E+08 | 2.56E+11 | 3.68E+11 | 8.39E+11 | 3.19E+12 | 0.46 |
| 12 | Henan | 6.30E+08 | 3.92E+08 | 1.88E+09 | 1.02E+12 | 1.12E+12 | 2.56E+12 | 9.30E+12 | 0.51 |
| 13 | Hubei | 5.82E+08 | 3.92E+08 | 1.09E+09 | 6.09E+11 | 1.40E+12 | 1.30E+12 | 5.44E+12 | 0.61 |
| 14 | Hunan | 4.31E+08 | 5.83E+08 | 1.29E+09 | 7.86E+11 | 7.86E+11 | 1.36E+12 | 5.22E+12 | 0.56 |
| 15 | Jiangsu | 8.27E+08 | 5.98E+08 | 1.73E+09 | 1.67E+12 | 3.91E+12 | 2.72E+12 | 1.27E+13 | 0.65 |
| 16 | Jiangxi | 2.33E+08 | 4.20E+08 | 8.84E+08 | 5.35E+11 | 8.45E+11 | 3.85E+11 | 2.93E+12 | 0.60 |
| 17 | Jilin | 2.48E+08 | 9.70E+07 | 2.79E+08 | 2.60E+11 | 5.10E+11 | 1.04E+12 | 4.52E+12 | 0.40 |
| 18 | Liaoning | 4.35E+08 | 1.14E+08 | 5.86E+08 | 2.92E+11 | 1.07E+12 | 1.68E+12 | 6.82E+12 | 0.45 |
| 19 | Inner Mongolia | 2.01E+08 | 1.38E+08 | 2.60E+08 | 4.73E+11 | 5.94E+11 | 1.20E+12 | 5.39E+12 | 0.42 |
| 20 | Ningxia Hui | 6.27E+07 | 2.57E+07 | 7.80E+07 | 1.20E+11 | 5.62E+10 | 1.83E+11 | 8.53E+11 | 0.42 |
| 21 | Qinghai | 4.07E+07 | 2.56E+07 | 6.07E+07 | 1.26E+11 | 5.47E+10 | 8.76E+10 | 6.26E+11 | 0.43 |
| 22 | Shaanxi | 2.59E+08 | 2.42E+08 | 6.46E+08 | 7.19E+11 | 5.22E+11 | 9.62E+11 | 4.25E+12 | 0.52 |
| 23 | Shandong | 7.49E+08 | 5.27E+08 | 1.85E+08 | 1.74E+12 | 1.05E+12 | 3.23E+12 | 1.32E+13 | 0.46 |
| 24 | Shanghai | 4.70E+08 | 1.10E+08 | 1.68E+08 | 1.99E+12 | 3.68E+11 | 3.92E+11 | 4.57E+12 | 0.60 |




| | | | | | | | | | |
|---|---|---|---|---|---|---|---|---|---|
| 25 | Shanxi | 2.53E+08 | 2.06E+08 | 4.86E+08 | 6.58E+11 | 3.61E+11 | 5.89E+11 | 3.27E+12 | 0.49 |
| 26 | Sichuan | 4.72E+08 | 5.51E+08 | 1.76E+09 | 7.95E+11 | 7.67E+11 | 1.81E+12 | 5.77E+12 | 0.58 |
| 27 | Tianjin | 2.19E+08 | 4.45E+07 | 1.18E+08 | 1.43E+12 | 1.90E+11 | 3.27E+11 | 3.88E+12 | 0.50 |
| 28 | Xinjiang | 1.81E+08 | 8.60E+07 | 3.10E+08 | 5.37E+11 | 1.96E+11 | 2.92E+11 | 2.19E+12 | 0.47 |
| 29 | Tibet | 8.73E+06 | 1.22E+07 | 6.92E+07 | 2.41E+10 | 3.06E+10 | 8.57E+10 | 3.37E+11 | 0.42 |
| 30 | Yunnan | 1.77E+08 | 2.33E+08 | 8.66E+08 | 2.83E+11 | 3.30E+11 | 8.14E+11 | 3.27E+12 | 0.44 |
| 31 | Zhejiang | 4.56E+08 | 4.10E+08 | 1.59E+09 | 1.20E+12 | 8.98E+11 | 2.84E+12 | 7.80E+12 | 0.63 |
| | In total: | 1.06E+10 | 8.04E+09 | 2.40E+10 | 2.89E+13 | 1.49E+13 | 3.38E+13 | 1.48E+14 | 0.53 |

Note: (a) In this study, for each of the 17 building sub-types in each grid of urban/township/rural level in each province/municipality, the same unit construction price is used; (b) The modelled floor area and replacement value in this study are particularly for residential buildings; (c) The net capital stock value estimated in Wu et al. (2014) refers to the depreciated asset value of residential, non-residential buildings, and infrastructure as well; (d) The building construction price used in this study and that in Wu et al. (2014) are not equal.



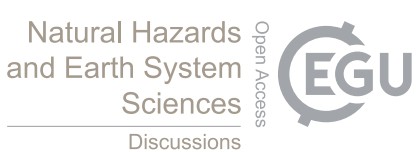
**Table 5:** Comparison of modelled floor area with Shanghai Statistical Yearbook records in 2015 (in this table, scientific notation is used to represent the large numbers).

| District | 2015 GHS (pop1) | pop1 (%) | Initially modelled population and floor area (km²) | | | | | | Adjustment factor | | Adjusted floor area (km²) | 2015 Shanghai Statistical Yearbook records | | | |
| --- | --- | --- | --- | --- | --- | --- | --- | --- | --- | --- | --- | --- | --- | --- | --- |
| | | | 1 | 2-3 | 4-6 | 7-9 | ≥10 | Sum of floor area in all storey classes | UCC | De-amp. factor | | floor area difference (%) | residential floor area (km²) | Recorded population (pop2) | pop2 (%) |
| Baoshan | 2.70E+06 | 10% | 12.7 | 23.6 | 33.5 | 2.8 | 10.8 | 83.3 | 0.83 | | 52.4 | -3% | 54 | 2.02E+06 | 8% |
| Chongming | 6.67E+05 | 3% | 7.0 | 12.6 | 1.4 | 0.0 | 0.2 | 21.3 | 0.74 | | 12.0 | 23% | 9.7 | 7.02E+05 | 3% |
| Fengxian | 1.26E+06 | 5% | 8.6 | 15.8 | 10.9 | 0.8 | 3.2 | 39.4 | 0.85 | | 25.2 | 17% | 21.6 | 1.17E+06 | 5% |
| Jiading | 2.49E+06 | 9% | 12.1 | 22.2 | 30.2 | 2.5 | 9.7 | 76.7 | 1.03 | | 60.0 | 78% | 33.6 | 1.57E+06 | 6% |
| Jinshan | 8.54E+05 | 3% | 5.7 | 10.5 | 7.8 | 0.5 | 2.2 | 26.7 | 0.91 | | 18.4 | 25% | 14.7 | 7.97E+05 | 3% |
| Minhang | 4.20E+06 | 16% | 15.3 | 29.0 | 58.9 | 5.1 | 19.6 | 127.9 | 0.92 | | 88.6 | 18% | 75.2 | 2.54E+06 | 10% |
| Pudong | 4.74E+06 | 18% | 27.6 | 51.1 | 49.5 | 3.9 | 15.2 | 147.2 | 1.11 | | 124.0 | -13% | 142.9 | 5.45E+06 | 22% |
| Qingpu | 1.33E+06 | 5% | 8.5 | 15.6 | 12.5 | 0.9 | 3.6 | 41.1 | 0.90 | 1.32 | 27.9 | 32% | 21.2 | 1.21E+06 | 5% |
| Songjiang | 1.96E+06 | 7% | 9.7 | 18.0 | 23.4 | 1.9 | 7.4 | 60.5 | 0.84 | | 38.2 | -3% | 39.5 | 1.76E+06 | 7% |
| **Downtown** | 5.99E+06 | 23% | 22.9 | 43.3 | 82.5 | 7.2 | 27.5 | 183.4 | 1.06 | | 147.3 | -26% | 198.5 | 7.05E+06 | 29% |
| | Sum: 2.62E+07 | | | | | | | Sum: 807.5 | | | | Sum: -3% | Sum: 610.9 | Sum: 2.43E+07 | |

Note: In "Adjustment factor" column, "**UCC**" is the abbreviation of Uniform Construction Price, derived in Table 7 and used to adjust the development disparity in districts of Shanghai; "**De-amp. factor**" is the averaged de-amplification factor, derived in Table 6 and used to adjust the amplification of population from 2010 census to 2015 GHS population; "**Downtown**" area includes eight administrative districts of Shanghai, namely Yangpu, Hongkou, Zhabei, Putuo, Changning, Xuhui, Jing'an and Huangpu.

**Table 6:** Derivation process of the de-amplification factor "1.32" in Table 5.

| Shanghai urbanity | Modelled floor area (km²), without adjustment | Ratio (%) | Amp. Factor from 2010 census to 2015 GHS population | De-amp. factor |
| --- | --- | --- | --- | --- |
| 1024 | 469.6 | 63% | 1.33 | |
| 2024 | 110.4 | 15% | 1.34 | **1.32** |
| 3024 | 167.8 | 22% | 1.29 | |





**Table 7**: Derivation of Uniform Construction Cost (UCC) in Table 5 from Shanghai 2015 Statistical Yearbook, to reflect the development disparity among districts of Shanghai (in this table, scientific notation is used to represent the large numbers).

| District | Population | District GDP | GDP/capita | UCC |
|---|---|---|---|---|
| Baoshan | 202400 | 1.10E+11 | 54147 | 0.83 |
| Chongming | 70160 | 2.72E+10 | 38791 | 0.74 |
| Fengxian | 116760 | 6.68E+10 | 57246 | 0.85 |
| Jiading | 156620 | 1.63E+11 | 104084 | 1.03 |
| Jinshan | 79710 | 5.70E+10 | 71509 | 0.91 |
| Minhang | 253950 | 1.84E+11 | 72603 | 0.92 |
| Pudong | 545120 | 7.11E+11 | 130414 | 1.11 |
| Qingpu | 120830 | 8.27E+10 | 68476 | 0.90 |
| Songjiang | 175590 | 9.69E+10 | 55212 | 0.84 |
| Downtown | 704540 | 7.96E+11 | 113012 | 1.06 |
| | sum: 2425680 | sum: 2.29E+12 | average: 94607 | |

Note: "Downtown" area includes eight districts of Shanghai located in the downtown area, namely Yangpu, Hongkou, Zhabei, Putuo, Changning, Xuhui, Jingan and Huangpu.




## Appendix

**Table A1**: The population thresholds used to divide the grids in 2015 GHS population density profile into urban/township/rural level.

| Province | Province ID | Population of each urbanity level in 2010-census | | | | Population share (%) | | | Population threshold (PT) | |
| --- | --- | --- | --- | --- | --- | --- | --- | --- | --- | --- |
| | | urban | township | rural | sum | urban | township | rural | PT1 (urban/township) | PT2 (township/rural) |
| Anhui | 1 | 12182587 | 13394530 | 33923351 | 59500468 | 20.47% | 22.51% | 57.01% | 13991 | 6908 |
| Beijing | 2 | 15563215 | 1295477 | 2753676 | 19612368 | 79.35% | 6.61% | 14.04% | 2709 | 1784 |
| Chongqing | 3 | 8681611 | 6614192 | 13550367 | 28846170 | 30.10% | 22.93% | 46.97% | 11194 | 5415 |
| Fujian | 4 | 12548384 | 8513556 | 15832277 | 36894217 | 34.01% | 23.08% | 42.91% | 6177 | 2621 |
| Gansu | 5 | 5258935 | 3932250 | 16384078 | 25575263 | 20.56% | 15.38% | 64.06% | 15175 | 9350 |
| Guangdong | 6 | 52383382 | 16641873 | 35290204 | 104320459 | 50.22% | 15.95% | 33.83% | 4427 | 2521 |
| Guangxi | 7 | 8352777 | 10065066 | 27605918 | 46023761 | 18.15% | 21.87% | 59.98% | 11711 | 5087 |
| Guizhou | 8 | 5537562 | 6199971 | 23011023 | 34748556 | 15.94% | 17.84% | 66.22% | 18126 | 10384 |
| Hainan | 9 | 2324288 | 1984228 | 4362969 | 8671485 | 26.80% | 22.88% | 50.31% | 8098 | 3658 |
| Hebei | 10 | 14388021 | 17187307 | 40278882 | 71854210 | 20.02% | 23.92% | 56.06% | 5670 | 2402 |
| Heilongjiang | 11 | 14122516 | 7201199 | 16990276 | 38313991 | 36.86% | 18.80% | 44.34% | 3845 | 1483 |
| Henan | 12 | 18331493 | 17888274 | 57810172 | 94029939 | 19.50% | 19.02% | 61.48% | 15203 | 8451 |
| Hubei | 13 | 17928160 | 10516925 | 28792642 | 57237727 | 31.32% | 18.37% | 50.30% | 11667 | 6345 |
| Hunan | 14 | 12738442 | 15714621 | 37247699 | 65700762 | 19.39% | 23.92% | 56.69% | 13563 | 5881 |
| Jiangsu | 15 | 30166466 | 17205022 | 31289453 | 78660941 | 38.35% | 21.87% | 39.78% | 6554 | 3341 |
| Jiangxi | 16 | 7504291 | 11995669 | 25067837 | 44567797 | 16.84% | 26.92% | 56.25% | 11309 | 3403 |
| Jilin | 17 | 10196745 | 4451454 | 12804616 | 27452815 | 37.14% | 16.21% | 46.64% | 6150 | 2849 |
| Liaoning | 18 | 22021184 | 5166779 | 16558360 | 43746323 | 50.34% | 11.81% | 37.85% | 3486 | 1874 |
| Inner Mongolia | 19 | 8011564 | 5708610 | 10986117 | 24706291 | 32.43% | 23.11% | 44.47% | 11152 | 5041 |
| Ningxia | 20 | 2050295 | 962727 | 3279328 | 6301350 | 32.68% | 15.28% | 52.04% | 11659 | 7624 |
| Qinghai | 21 | 1368033 | 1148221 | 3110469 | 5626723 | 24.31% | 20.41% | 55.28% | 11850 | 5088 |
| Shaanxi | 22 | 8837175 | 8222162 | 20268042 | 37327379 | 23.67% | 22.03% | 54.30% | 13716 | 6862 |
| Shandong | 23 | 28364984 | 19255743 | 48171992 | 95792719 | 29.61% | 20.10% | 50.29% | 6587 | 3373 |
| Shanghai | 24 | 17640842 | 2914256 | 2464098 | 23019196 | 76.64% | 12.66% | 10.70% | 4927 | 2736 |
| Shanxi | 25 | 9414053 | 7746486 | 18551562 | 35712101 | 26.36% | 21.69% | 51.95% | 8763 | 3873 |
| Sichuan | 26 | 15915660 | 16428768 | 48073100 | 80417528 | 19.79% | 20.43% | 59.78% | 14668 | 8133 |
| Tianjin | 27 | 8858126 | 1419767 | 2660800 | 12938693 | 68.46% | 10.97% | 20.56% | 3141 | 1868 |
| Xinjiang | 28 | 6071803 | 3263949 | 12480063 | 21815815 | 27.83% | 14.96% | 57.21% | 10473 | 3618 |
| Tibet | 29 | 272322 | 408267 | 2321576 | 3002165 | 9.07% | 13.60% | 77.33% | 9751 | 4483 |
| Yunnan | 30 | 6324830 | 9634242 | 30007694 | 45966766 | 13.76% | 20.96% | 65.28% | 18128 | 8818 |
| Zhejiang | 31 | 20386294 | 13163915 | 20876682 | 54426891 | 37.46% | 24.19% | 38.36% | 5594 | 2504 |

Note: For each province, **PT1** and **PT2** are two population thresholds used to assign the grids in 2015 GHS population density profile with urban, township and rural attributes, according to the population density $\lambda$ in each grid with 1km×1km resolution. The detailed criteria are that: if $\lambda \geq PT1$, the grid is assigned as **urban**; if $PT1 > \lambda \geq PT2$, **township**; if $\lambda < PT2$, **rural**.