# Peer review of "Residential building stock modelling for mainland China"

_Natural Hazards and Earth System Sciences, 2019_

## Referee Comment (RC1) · Anonymous Referee #1 · 12 Jun 2020

To whom it may concern,

I greatly appreciate the authors' efforts to address the issue of "grid-scale building stock modeling", which is very helpful for the understanding of spatial distributions of seismic risk and beyond. However, I feel the manuscript contains some major problems, rendering it inappropriate for publication at the current version. Normally, a manuscript in the state and quality like this one should be rejected. But, I feel the overall modeling idea and process presented in the manuscript is generally reasonable and acceptable, and part of the problems might be due to language usages or writing experience. Thus, I think the paper has the potential and the possibility to be a valuable contribution to the disaster research community if it undergoes a very careful and intensive revision process. I am therefore still willing to suggest the Journal rating it as major revision.

[Figure]

I strongly recommend the authors making a most careful, detailed, and thorough revision to make sure the manuscript's scientific soundness and expression readability. Below, please see my general and detailed comments:

General comments:

1) Some key elements in the modeling process (including validation section) haven't been clearly explained or presented, making readers have to guess (some of them I can figure out, others not) what they are specifically, how they are produced, and why using them is reasonable, such as F2, F3, 9 equations and others in the modeling process, and UCC, "1.32" in the validation section. I will try to specify them one by one in the next part "specific comments part".

2) It is difficult for me to believe that the validation section is convincing. (a) First, the central theme of this effort is to disaggregate administrative unit resolution building stock data into "km grid" resolution counterpart. Thus for me, the logically nature and common validation should be verifying to what extent "this study' s result: km grid building stock distribution" is in line with "the actual building stock in those km grids". However, the paper used two kinds of comparisons in provincial and district levels respectively to validate the reliability of the modeling. For me, this seems something like "self validates self" in "scale" sense, in other words, one "large scale" things verifies another things with approximately same large-scale, which seems to not meet with the paper's central theme: getting large scale things smaller ones. Here, it is worthy to be considered further what is the nature or the essence of the two comparisons/validations the paper provided? (b) Second, in section "3.2.1: Provincial-level based comparison between...", there is "the value of (A)/(C) varies within the range of 0.31-0.65, which indicates the high consistency..." why and how you concluded this? This is really confused me. If we calculate another kind of ratios, for example, the ratios between this study's results (after getting them provincial level) with each province's population numbers in the current yearbooks, we perhaps could also get a series of ratios with small fluctuations (I am not sure, but worthy of try)... (c) Third, for section "3.2.2:

District-level based comparison between. . ." First, how about other provinces? Second, and more importantly, I feel it might be better to put the two adjustment steps into the whole modeling process (i.e., making them as part of the modeling itself, namely part of Fig1, not regarding/treating them as validations), please consider.

3) Large part of the introduction (esp. the first half part) and the limitation discussion section are both too general. The overarching and/or specific objectives (e.g., why making efforts to address building stock's financial/economic value), significance, application prospects, and if the planned objectives were delivered haven't been clearly communicated, which should be relevant in part to the lack of the necessary introduction of other methods that also address building-stock, such as ATC-series (esp.13) of USA, EMS98 of Europe, ATC series-based variants in China, various remote sensing based or associated methods, and so on.

4) Please change the term "construction material" into "structure type" in the corresponding places of the whole manuscript, so as to meet the relevant conventional concept/term of the earthquake engineering field. Please make it clear what does the phrase "residential building stock valve" specifically refer to, in other words, please make this value-focused phrase more accurate (i.e., what value? floor area or others?). It should be the financial or economic value of residential building stock in this paper, so please use a proper wording of this meaning consistently in the whole manuscript.

5) there are a good number of statement accuracy, nuance, or language wording/expression problems in the whole manuscript, lowering greatly its readability and making its contents sometimes very hard to follow. I will try to specify them in the next section "specific comments section", but not all. I strongly recommend authors paying enough attention to this aspect.

Specific comments:

Line 1 (of the PDF doc. of the original manuscript, hereinafter): the paper's title is too general, which cannot convey clearly the central theme of the work.

Line 11: regarding "……especially in developing countries". I don't think there is a need to supplement this general but vague emphasis, which might make readers wonder "are there other stories (i.e., building damages/collapses are not the leading cause......) in the developed world? Actually, for the long-established saying "Earthquakes Don't Kill People, Buildings Do", it is the same everywhere.

Line11-12: regarding"……targeted at near-real time post-earthquake mitigation". I cannot understand why this info. is emphasized (esp., in the abstract), I also cannot figure out well how specifically risk analysis can contribute to this stage. Especially, the whole manuscript doesn't contain any specific explanation, discussion or connections on this at all.

Line 14: for me, "using population density profile as the proxy" reads awkward, "using population density profile as a bridge" might be better (for the whole manuscript, the same or similar below/hereinafter). In addition, for clarity, accuracy, and information completeness (e.g., using what to do what), "…by disaggregating relevant urbanity level data in the 2010- census of each province into km grid scale and using population density profile provided in 2015 GHS as a bridge", or similar expression like this might be preferable. Please check and consider.

Line 10-24: I feel the whole abstract need to be re-generalized after a thorough major revision of the whole manuscript.

Between 24 and 25: commonly, keywords should be provided.

Line 26-28: (1) "being" vs. "Target B" reads awkward; changing "being" into "including" might be better. (2) "IDDR 2018" and "over years" are inconsistent in meaning (i.e., from 2018 to 2019, there is only 1 year……. (3)further, what is the relationship between the main and the subordinate clauses here?

Line 44: "As such" reads awkward, please check.

Line 48: There are provincial level, prefecture-level, district-level, and grid-level things

that were addressed or mentioned in the whole paper. My wonder is what does the term "country-level" here specifically mean? You mean addressing/treating something in a country as a whole or addressing them/it in a whole country (country-wide)? Please check, and use precise expressions. Do please avoid using the general or vague wording as such in the whole manuscript; and there are quite many of them. Do please pay enough attention to this.

Line 50: (1) please change "construction age and material" into "age and structure type" (for the whole manuscript, the same or similar below/hereinafter). (2) "are used" reads awkward, please check. "can be used"?

Line 64: "in turn" reads awkward. Please check.

Line 25-87(i.e., the whole introduction part): (1) commonly, the specific objective of the work should be clearly communicated at the very end of the introduction. Unfortunately, the current manuscript doesn't provide this. Do please add. (2) a large part of the introduction (esp. the first half part) is too general, and even not directly relevant to the central theme of the paper. Especially, I don't think there is a need to communicate those kinds of basic knowledge in an academic paper, such as what is hazard, exposure, and vulnerability (and frankly, I feel some of the existing expressions of these terms read not that accurate). Instead, I feel other methods that also address building-stock, and advantages and disadvantages of them should be succinctly discussed, including ATC-series (esp.13) of USA, EMS98 of Europe, ATC series-based variants in China, various remote sensing-based or associated methods, and so on, which should be a great help for the authors to refine the specific objectives of their current work. Do please consider this.

Line 97: Please change "construction material" into "structure type".

Line 105-106: the current sentence "one advantage of the 2010-census data is its further categorization of data into three urbanity levels, which better reflects the regional difference within each province" is inaccurate and even not reasonable; because, for a

given province in the 2010-census, the associated urbanity levels were only provided for the province as a whole, there aren't spatial distribution info.

Line 124-125: Please make the meaning of the sentence ". . .before disaggregating the urbanity-level based census data into each grid" clearer, i.e., make it clear that you will disaggregate which set of data; and it might be good to revise the sentence as "before disaggregating the urbanity-level based data in 2010 census into each grid". Do please avoid such kind of vague or incomplete expressions in the whole manuscript, so as to make the text easy to follow.

Line 128-129: I really cannot understand what is the relationship between "Aubrecht et al. (2015) and Gunasekera et al. (2015)'s approach (although I did read these two papers)" and "the urban/township/rural population proportions of each province in 2010-census data set"; aren't these proportions provided in the 2010-census directly? Or cannot these proportions be easily calculated from info. in the 2010-census directly? Or, did I misunderstand your original intended meaning here? i.e., when talking about "the urban/township/rural population proportion of each province" here, you refer to those of the 2015 GHS data set, right? So, do please try to make the sentence of such kind as accurate as possible, so as to avoid getting readers lost.

Line 130-131: "the population proportion of urban/township/rural urbanity level is 76.64%, 12.66% and 10.7%, respectively", which means that the population proportion of urban/township/rural urbanity levels "in the 2010-census" are 76.64%, 12.66% and 10.7%, respectively", right? If so, please make this info. complete and clear.

Line 131-132: The sentence "Then the grids (1km×1km) in 2015 GHS population density file of Shanghai are sorted from the largest to the smallest" reads awkward. Is it better if changing it into "Then the grids (1km×1km) of Shanghai in 2015 GHS file are sorted from the largest to the smallest in population density"? Please check.

Line 128-144: I guess there is an important assumption here, namely, the larger the population density, the higher the urbanized extent. If so, please write this out clearly

to avoid making readers have to guess. Please check and revise.

Line 163: "up to now" reads awkward, please check.

Line 165: "from the 2010-census" reads awkward, please check.

Line 169-170: I don't think "F2" has been explained clearly, including how they were calculated and how they were used subsequently in the amplification of the 2010-census data. I can guess these. But I think it is necessary to introduce them clearly in the text (e.g., using one example), so as to avoid making the readers have to make that guess.

Line 179-180: Similarly to "F2", the wording for "F3" is also vague and incomplete. Please check and revise.

Line 183-185: there is "the population in each grid living in building types grouped by number of storey (1, 2-3, 4-6, 7-9, $\geq$10) or by construction material (steel/RC, mixed, other, brick/wood) can be derived". It is hard to guess how you achieve this. I guess there is another very important assumption here. Specifically, from the 2010-census, we can get a series of provincial level percentages of the population living in buildings with different floors or with different structure types with one urbanity level (urban, township, or rural); then it is assumed that all the grids with this same urbanity level are all evenly/uniformly have these same percentages. My guess may be correct, may not. But the author should make this highly generalized statement clear enough, so as to make this key modeling step easier to be understood.

Line 186: "the number of buildings" reads awkward relative to the main topic of this paper. Please check and revise.

Line 201: "currently" and "for instance" both read awkward. Please check and revise.

Line 205-206: there is "9 equations". My wonder includes what are they and how they function specifically? Please explain them in detail. There is the phrase of "linear problem". why you suddenly say this, why the problem is linear, how this linear problem looks like?

Line 210-223: this part is very hard to follow. Please check and provide necessary details and explanations. For example, (1) in Line 216-219, there is "the remaining steel/RC buildings are proportioned to other storey classes from highest to lowest", and the like. Please specify, how you get these proportions? (2) regarding step 6 here, I think it is simply that the remaining buildings in each storey class are all belonging to "mixed" buildings.

Line 226: this sub-title cannot meet with the contents below.

Line 253-368: i.e., the two validations. Specific comments, please see the second point in the general comments section above.

Line 306-307: "on the other hand" reads awkward. Please check and revise. However?

Line 330: I feel more information regarding "UCC" should be provided, so that, it is easier for readers to understand why using UCC can make that adjustment. (existing studies of other researchers might have discussed this, but in the interest of the common requirement that a single paper had better be self-standing, key info. should be introduced)

Line 347-350: I feel the explanation of this "de-amplification factor" is not clear enough, more info. is needed. For example, I found that "1.32" is exactly the arithmetic mean of "1.33, 1.34 and 1.29". Is this just a coincidence? Please check and revise.

Line 414-420: I feel the current limitation discussion is too general. Instead, I feel the most relevant and direct limitation discussion (disadvantages and future improvement directions) should focus on those assumptions and "factors" that this modeling process used.

---

## Referee Comment (RC2) · Anonymous Referee #2 · 20 Jun 2020

The Introduction starts with some common sentences on earthquake loss and related action on the UN level, however, this paragraph is not very suitable to introduce the topic of the manuscript to the potential readers. Therefore, I kindly recommend re-writing. Moreover, the statement of world-wide earthquake loss should go clearly beyond the referred two studies of one of the co-authors of this manuscript. One more detail: For me it is not clear why in a paper from 2011 earthquake loss can be given in 2016 values, please clarify. Then the authors address the need to have information on the building stock level when risk assessment should be undertaken. The state that in cases where such information is not available, obtaining necessary information is not practicable. I strongly disagree with this statement, throughout the relevant literature there are many different methods presented of how to do so. This needs thorough

revision. Moreover, the authors elaborate on a method to compile such information by taking census data in consideration. How the building stock value is correlating to statistical information on population density (lines 49-57)? How did the authors generally treat the MAUP issue when using data bond to administrative borders? Further down the text body, the authors correctly state that "to better cope with this spatial mismatch between natural hazards [spatial occurrence] and administrative boundaries, building stock models should be geocoded with relatively high resolution and be disaggregated from more detailed census data". The last statement means that from a methodological point of view, such an assessment will not guide us to precise results that can be used as a proxy for the building stock. So somehow, the introductory section is unclear with respect to what the authors would like to show us in their study. Finally, the research gap is not properly defined, nor is the niche to be filled by this work easily accessible to potential readers.

In the method section the authors explain how they aggregated information on the building characteristics to information on the population density, both at a final resolution of 1 km grid cells. In this respect it remains unclear how the other building types were excluded from the grids, as information on e.g. building design and material in the statistics ("Long Table") are also related to other building types, right? -> Needs clarification.

Further on, the authors present different methods of how to merge different types of information such as the amount of buildings of different height or different construction type to these grid cells, resulting in a certain spatial probability for the different data. It remains open, however, how this information was finally be checked against the real world situation, and as such it remains open how e.g. information on population was distributed or allocated to different building categories. Occupation rate and building values were then allocated to the different building sub-categories, and spatially distributed over the grid cells.

Results of values per grid cell where then compared to (A) a study published by Wu

et al. (2014) on the net capital stock, (B) more detailed information available on the residential floor area for the Shanghai district, and (C) an empirical earthquake vulnerability study published by one of the co-authors of this manuscript, linking vulnerability to reported loss. The authors conclude that the results from the present manuscript (in terms of what? Potential value of buildings? Potential loss resulting from an earthquake scenario?) are in line with results from other studies, a statement which cannot be supported by the referee evaluating the information provided by the authors. In the present form, the results of the study are not validated, they are only opposed to other studies on building values (in case A), to the area used for computation of values (in case B) and to vulnerability, linking the newly generated building values to an empirical vulnerability function and comparing the results to some loss reports available (in case C). With respect to the latter, further questions arise with respect to different construction types and their individual structural vulnerability concerning earthquakes, this needs careful interpretation and more information on the comparison performed. As such, the added value of the material presented here is not clear to me. Statements such as "Therefore, the estimated loss range, based on the buildings stock model developed in this study and the empirical loss function developed in Daniell (2014), is quite compatible with that given in previous studies. This compatibility further validates the robustness of our residential building stock model" (lines 392ff.) may be right but cannot be proven from what the authors have shown.

Finally, information given in the conclusion is by far too generalized, instead, the overall limitation and the specific ones, such as those arising from the use of aggregated statistical data and the underlying factors, should be discussed in detail.

Hence, I cannot recommend publication of the work in its present form, the manuscript needs major revisions in terms of (1) problem statement, including extended literature review, (2) explanation of methods, (3) compilation of results, and – most important – (4) discussion, including discussion on limitations and uncertainties, so that (5) sound conclusions will become possible.

---

## Author Comment (AC1) · 30 Jul 2020

To whom it may concern,

I greatly appreciate the authors' efforts to address the issue of "grid-scale building stock modeling", which is very helpful for the understanding of spatial distributions of seismic risk and beyond. However, I feel the manuscript contains some major problems, rendering it inappropriate for publication at the current version. Normally, a manuscript in the state and quality like this one should be rejected. But, I feel the overall modeling idea and process presented in the manuscript is generally reasonable and acceptable, and part of the problems might be due to language usages or writing experience. Thus, I think the paper has the potential and the possibility to be a valuable contribution to the disaster research community if it undergoes a very careful and intensive revision process. I am therefore still willing to suggest the Journal rating it as major revision.

I strongly recommend the authors making a most careful, detailed, and thorough revision to make sure the manuscript's scientific soundness and expression readability. Below, please see my general and detailed comments:

General comments:
1) Some key elements in the modeling process (including validation section) haven't been clearly explained or presented, making readers have to guess (some of them I can figure out, others not) what they are specifically, how they are produced, and why using them is reasonable, such as F2, F3, 9 equations and others in the modeling process, and UCC, "1.32" in the validation section. I will try to specify them one by one in the next part "specific comments part".
Response: Thank you for your time spent on reviewing this paper and your constructive comments. Detailed responses to these comments are given as follows.

2) It is difficult for me to believe that the validation section is convincing.
(a) First, the central theme of this effort is to disaggregate administrative unit resolution building stock data into "km grid" resolution counterpart. Thus for me, the logically nature and common validation should be verifying to what extent "this study' s result: km grid building stock distribution" is in line with "the actual building stock in those km grids". However, the paper used two kinds of comparisons in provincial and district levels respectively to validate the reliability of the modeling. For me, this seems something like "self validates self" in "scale" sense, in other words, one "large scale" things verifies another things with approximately same large-scale, which seems to not meet with the paper's central theme: getting large scale things smaller ones. Here, it is worthy to be considered further what is the nature or the essence of the two comparisons/validations the paper provided?
Response: We totally understand your concern. It is true that since what we finally provide is gird-level residential building stock model (in terms of floor area and replacement value) for mainland China, the most convincing validation should also be at grid-level. However, considering the fact that the last time China conducted the country-wide building investigation

was in the 1985, which is 35 years ago, and for nowadays buildings no such in-situ survey data are currently available (Han et al., 2013). Such difficulty hindered the validation of modelled results in this study for the whole mainland China at grid level.

Although currently it is not practicable to valid our model for each grid, we do notice that in previous studies, there are such validation work performed for several regions in China. The first example is the study of Wu et al. (2019). They modelled the building asset replacement value for Shanghai by using the statistical building floor area (BFA) at district level, the population records at district level and township level, the building footprint map with 2.5m*2.5m resolution, and the LandScan2010 population density profile with 800m*800m resolution. In their modelling process, the district level building floor area was first disaggregated into township level by using the population ratio of each town as the apportion weight. Then, before they distributed township level BFA into the building footprint map grids with 2.5m*2.5m resolution by using the LandScan2010 population density profile as the bridge, the population in each LandScan2010 grid (800m*800m) was evenly distributed into building footprint map grids (2.5m*2.5m) within the same LandScan 2010 grid geographically. Finally, to validate their modelled building floor area at each footprint map grid (2.5m*2.5m), they conducted the building height survey in the downtown area of Shanghai. The final comparisons were presented at grids with resolutions of 100m*100m, 200m*200m, 400m*400m, 800m*800m and 1600m*1600m. As expected, the larger the grid size used, the better the consistency between the modelled BFA and the actual BFA. However, since our modelling area is the whole mainland China, and if we only randomly select some local area to conduct the site survey like Wu et al. (2019) did, statistically it might be not significantly enough. Furthermore, in Wu et al. (2019), the buildings they considered include more occupancy types (i.e. residence, office, commerce etc.). Therefore, in their in-situ survey, they only need to count the actual height of buildings, but do not need to figure out their occupancy type. While in our work, the model is developed for residential buildings only due to the lack of detailed records in the 2010-census data for other building types.

Another example that conducted grid-level site survey is the work of Han et al. (2013). They developed the population and building distribution models for the Zhaotong City of Yunnan Province in China with 1km*1km resolution. Their dasymmetric population distribution model was developed based data from the Yunnan Province Earthquake Emergency Foundation Database, which includes detailed population census for each town/street, the network of roads (including high-speed way, and state/province/county/village road), the distribution of dwelling settlements, remote sensing land use data, digital elevation model etc. Land use and road network data were used to determine the weight to distribute the township/street level population into each 1km*1km grid based on regression analysis. The building distribution model was created based on the population model and other information derived from their in-situ survey of 240 sample girds. Such information includes the floor area per capita and the ratio of different building structure types in grids with different population density range.

Similar to the work of Han et al. (2013), by using the data for the 18 towns of Ning'er County of Yunnan Province from the same database as Han et al. (2013) , Chen et al., (2012) visually

interpreted the locations of 171 township dwelling settlements and 4703 rural dwelling settlements from remote sensing images of IRS-P5 (2.5m resolution) and RapidEye (5m resolution), and calculated the population density in each 1km*1km grid using regression analysis. Based on the methods in Chen et al. (2012) and Han et al. (2013), Xu et al. (2016) developed the 1km*1km population and building distribution maps by disaggregating county-level 2010-census data for whole mainland China, targeted for rapid and accurate fatality and financial loss assessment after damaging earthquake occurs. The outputs in Xu et al. (2016) have been officially used to provide post-earthquake rescue suggestions for the government in the 2017 Ms7.0 Jiuzhaigou earthquake. While, due to our limited access to data and in-situ survey results for Yunnan Province in Chen et al. (2012) and Han et al. (2013), and the detailed foundation database for the whole mainland China used in Xu et al. (2016), we cannot conduct a more detailed comparison to validate our model. Furthermore, although the outputs of Han et al. (2013) and Xu et al. (2016) included building distribution map, they did not differentiate the occupancy type of buildings (i.e. residential, commercial, industrial etc.), we thus cannot compare our results with their outputs directly.

You are right that our paper's central theme is to "get large scale things smaller ones" and this is to better reflect the spatial heterogeneity of residential building stock distribution, compared to the average distribution of census data within an administrative area. For example, after the occurrence of 2008 Wenchuan Ms8.0 earthquake, one of the most severely affected areas, Qingchuan County, did not get an appropriate and in-time rescue response, while most the rescue materials were sent to the less damaged Dujiangyan City. One of the reasons for this problem is that the disaster exposure data (population, buildings) are based on administrative units (Xu et al., 2016).

When it comes to the validation of our modelled building stock replacement value and floor area, a reasonable test is to check whether the model can be used to better reflect the seismic loss distribution pattern or to compare modelled results with outputs derived by using different methods at coarser level. And we do not think this is a "self validates self" thing, since when disaggregating the urbanity level census data into grid level, this process is not related to or affected by prefecture or county/district boundary. And we think if we reaggregate the grid-level data into county/district/prefecture level and compare them with other independent sources of validation data. If they are still consistent, then it can be regarded as a positive signal indicating the reliability of our modelled results.

The three validation tests in this work are performed for different purposes. The first one is to compare our modelled residential building replacement value with the net capital stock value in Wu et al. (2014) at provincial level, to have a general check of whether the building replacement value we approximate is of reasonable magnitude. We then compare our modelled residential building floor area at district level with the statistical results of Shanghai only, since such data in other provinces are not available. Finally, we apply our model to a scenario earthquake financial loss assessment, which is also necessary, since the motivation for us to conduct this work is exactly for seismic risk analysis. And if all three tests reflect the reliability of this model, then the work is meaningful. At least the fully reproducible results presented in

this study can be considered to be integrated into the ongoing international initiative such as the Global Earthquake Model (GEM), in which the exposure data for China currently is of 5km*5km resolution and disaggregated from provincial census data.

(b) Second, in section "3.2.1: Provincial-level based comparison between…", there is "the value of (A)/(C) varies within the range of 0.31-0.65, which indicates the high consistency…" why and how you concluded this? This is really confused me. If we calculate another kind of ratios, for example, the ratios between this study's results (after getting them provincial level) with each province's population numbers in the current yearbooks, we perhaps could also get a series of ratios with small fluctuations (I am not sure, but worthy of try)

Response: Our model and that of Wu et al. (2014) are different in method, model contents (we consider residential buildings, but they consider both residential and non-residential buildings) and the treatment of depreciation issue (we do not consider the depreciation and estimate the replacement value, but they considered this issue and estimated the net value). In spite of the above differences, for each province, our estimated residential building replacement value is around 0.31~0.65 of the net value of all building types estimated in Wu et al. (2014). Recently, we also checked the average ratio of value between newly constructed residential building and the value of all newly constructed buildings in the past five years. And the average ratio for 31 provinces is within the range of 0.38~0.75. Therefore, we think the ratio between our result and that of Wu et al. (2014) is reasonable. By doing this comparison, we were not to emphasize the small fluctuation of this ratio range, but to make that the residential building replacement value we approximate is of reasonable magnitude.

(c) Third, for section "3.2.2: District-level based comparison between…" First, how about other provinces? Second, and more importantly, I feel it might be better to put the two adjustment steps into the whole modeling process (i.e., making them as part of the modeling itself, namely part of Fig1, not regarding/treating them as validations), please consider.

Response: Thank you for this suggestion. After a broader check of databases related to building statistical data, we found that although in other provinces there are no corresponding records of accumulated residential building floor area like that given in Shanghai 2015 statistical yearbook, there are subsets of 2010-cenus data, (although not with open access, but can be bought), in which the residential building floor area are aggregated at county/district level for each province. Since when disaggregating the urbanity level census data into grid level, this process is not related to or affected by prefecture or county/district boundary, now we find the census records aggregated exactly at county/district level and it will be very meaningful to compared our modelled results with census records at county/district level. This comparison will be added in the revised version of the manuscript.

For the second suggestion in this comment, we think whether the adjustment by using UCC and de-amplification factor needs to be conducted should be decided after comparing the modelled results with the actual situation in specific research area. Due to the huge territory of China, the economic disparity and geographic climatic diversity are widely spanned and the standardization in building construction also varies in different periods. In an ideal case, the dasymmetric building distribution model should also be defined separately. However, it would

be too complicated to build a unique model for each region specifically. That is why we use the same modelling process for all the 31 provinces. We also point out that to improve accuracy in future seismic risk analysis, our modelled results should be adjusted according to the actual situation in the study area, like we tried for Shanghai. But, such adjustment may not always be necessary for every province. Therefore, we did not apply such adjustment in the modelling process.

To provide convenience for such need to adjust our modelled results like we do for Shanghai in the future, we will upload the value of UCC index and the de-amplification factor for all counties in mainland China. The UCC index is derived based on the statistical data of GDP in 2010 and the de-amplification factor is derived based on our modelled floor area for each urbanity level of each province and the amplification we used to scale the 2010-census population to 2015 GHS population.

3) Large part of the introduction (esp. the first half part) and the limitation discussion section are both too general. The overarching and/or specific objectives (e.g., why making efforts to address building stock's financial/economic value), significance, application prospects, and if the planned objectives were delivered haven't been clearly communicated, which should be relevant in part to the lack of the necessary introduction of other methods that also address building-stock, such as ATC-series (esp.13) of USA, EMS98 of Europe, ATC series-based variants in China, various remote sensing based or associated methods, and so on.
Response: Thank you for this suggestion. We did describe the motivation of this work, its significance and application prospects in Page 2 Line 42-48, which is probably kind of general. We will enrich this part in the revised manuscript.

To have a more comprehensive understanding of other methods in exposure modeling as well as the use of remote sensing data in this area, we conduct a more extensive literature review. A succinct summarization of these literature will be given, to better locate the contribution of this work on the research map of this topic.

4) Please change the term "construction material" into "structure type" in the corresponding places of the whole manuscript, so as to meet the relevant conventional concept/term of the earthquake engineering field. Please make it clear what does the phrase "residential building stock value" specifically refer to, in other words, please make this value-focused phrase more accurate (i.e., what value? floor area or others?). It should be the financial or economic value of residential building stock in this paper, so please use a proper wording of this meaning consistently in the whole manuscript.
Response: Accepted! We will replace "construction material" by "structure type". In this work, we modelled both the floor area and replacement value of residential building stock in mainland China. But here the "Residential building stock value" specifically refers to the replacement value of the building inventories, which is mentioned in Page 6 Line 190 and afterwards. We will give a clearer claim earlier in the introduction section.

5) there are a good number of statement accuracy, nuance, or language wording/expression

problems in the whole manuscript, lowering greatly its readability and making its contents sometimes very hard to follow. I will try to specify them in the next section "specific comments section", but not all. I strongly recommend authors paying enough attention to this aspect.

Response: Thank you for your patience in reading this manuscript. We will pay more attention to the accuracy of our description.

**Specific comments:**

Line 1 (of the PDF doc. of the original manuscript, hereinafter): the paper's title is too general, which cannot convey clearly the central theme of the work.

Response: To better reflect the contents of this work, the topic will be changed to "Census-based grid-level residential building stock modelling for seismic risk assessment in mainland China".

Line 11: regarding "……especially in developing countries". I don't think there is a need to supplement this general but vague emphasis, which might make readers wonder "are there other stories (i.e., building damages/collapses are not the leading cause......) in the developed world? Actually, for the long-established saying "Earthquakes Don't Kill People, Buildings Do", it is the same everywhere.

Response: Accepted. We will remove the description of "especially in developing countries".

Line11-12: regarding"……targeted at near-real time post-earthquake mitigation". I cannot understand why this info. is emphasized (esp., in the abstract), I also cannot figure out well how specifically risk analysis can contribute to this stage. Especially, the whole manuscript doesn't contain any specific explanation, discussion or connections on this at all.

Response: The initial consideration is that the exposure modelling is one important component of seismic risk analysis, which provides key references in resource allocation and dissemination for near-real time post-earthquake mitigation. But you are right that our focus is the modelling of residential building stock replacement value and floor area, not the link between this focus and post-earthquake mitigation. We will remove the description of "targeted at near-real time post-earthquake mitigation" in the abstract.

Line 14: for me, "using population density profile as the proxy" reads awkward, "using population density profile as a bridge" might be better (for the whole manuscript, the same or similar below/hereinafter). In addition, for clarity, accuracy, and information completeness (e.g., using what to do what), "…by disaggregating relevant urbanity level data in the 2010-census of each province into km grid scale and using population density profile provided in 2015 GHS as a bridge", or similar expression like this might be preferable. Please check and consider.

Response: Thank you for this suggestion. We will replace "using population density profile as the proxy" by "using population density profile as a bridge". And incomplete information will be specified after a thorough check.

Line 10-24: I feel the whole abstract need to be re-generalized after a thorough major revision of the whole manuscript.

Response: We will modify the abstract accordingly in the revised manuscript.

Between 24 and 25: commonly, keywords should be provided.
Response: Accepted. The key words "residential building stock, replacement value, 2010-census data, dasymmetric disaggregation" will be provided.

Line 26-28: (1) "being" vs. "Target B" reads awkward; changing "being" into "including" might be better. (2) "IDDR 2018" and "over years" are inconsistent in meaning (i.e., from 2018 to 2019, there is only 1 year… (3)further, what is the relationship between the main and the subordinate clauses here?
Response: Accepted. We will replace "being" by "including" and remove the description of "over the years". The description in the subordinate serves as a proof of the claim in the main clause.

Line 44: "As such" reads awkward, please check.
Response: Accepted. We will replace "As such" by "Therefore".

Line 48: There are provincial level, prefecture-level, district-level, and grid-level things that were addressed or mentioned in the whole paper. My wonder is what does the term "country-level" here specifically mean? You mean addressing/treating something in a country as a whole or addressing them/it in a whole country (country-wide)? Please check, and use precise expressions. Do please avoid using the general or vague wording as such in the whole manuscript; and there are quite many of them. Do please pay enough attention to this.
Response: Thank you. We will replace "country-level" by "country-wide".

Line 50: (1) please change "construction age and material" into "age and structure type" (for the whole manuscript, the same or similar below/hereinafter). (2) "are used" reads awkward, please check. "can be used"?
Response: Accepted. We will replace "construction age and material" by "age and structure type" in the whole context. And "are used" here will be replaced by "can be used".

Line 64: "in turn" reads awkward. Please check.
Response: Accepted. We will delete "in turn".

Line 25-87(i.e., the whole introduction part): (1) commonly, the specific objective of the work should be clearly communicated at the very end of the introduction. Unfortunately, the current manuscript doesn't provide this. Do please add. (2) a large part of the introduction (esp. the first half part) is too general, and even not directly relevant to the central theme of the paper. Especially, I don't think there is a need to communicate those kinds of basic knowledge in an academic paper, such as what is hazard, exposure, and vulnerability (and frankly, I feel some of the existing expressions of these terms read not that accurate). Instead, I feel other methods that also address building-stock, and advantages and disadvantages of them should be succinctly discussed, including ATC-series (esp.13) of USA, EMS98 of Europe, ATC series-based variants in China, various remote sensing-based or associated methods, and so on, which

should be a great help for the authors to refine the specific objectives of their current work. Do please consider this.

Response: Thank you. We will add the following description of the objective of this work in the end of the introduction: "The purpose of this work is to develop grid-level residential building stock model for seismic risk assessment in mainland China and this is achieved by disaggregating 2010-census into 1km*1km grid with the 2015 GHS population density profile as the bridge."

The introduction of hazard, exposure and vulnerability will be removed, and a succinct summarization of methods related to exposure modeling as well as the difference in building structure type recorded in the census of China and in ATC-13 will be mentioned.

Line 97: Please change "construction material" into "structure type".

Response: Accepted. All description of "construction material" will be replaced by "structure type".

Line 105-106: the current sentence "one advantage of the 2010-census data is its further categorization of data into three urbanity levels, which better reflects the regional difference within each province" is inaccurate and even not reasonable; because, for a given province in the 2010-census, the associated urbanity levels were only provided for the province as a whole, there aren't spatial distribution info.

Response: Thank you for pointing this out. This expression was to emphasize that the building related data in 2010-census are given for urban/township/rural level separately, which is an advantage compared with mixed them all together, otherwise additional assumptions need to be made to further separate them into different urbanity levels. We will replace the description of "which better reflects the regional difference within each province" by the above explanation.

Line 124-125: Please make the meaning of the sentence "…before disaggregating the urbanity-level based census data into each grid" clearer, i.e., make it clear that you will disaggregate which set of data; and it might be good to revise the sentence as "before disaggregating the urbanity-level based data in 2010 census into each grid". Do please avoid such kind of vague or incomplete expressions in the whole manuscript, so as to make the text easy to follow.

Response: Accepted. We will replace "before disaggregating the urbanity-level based census data into each grid" by "before disaggregating the urbanity-level based data in 2010-census into each grid". Other incomplete expressions will be checked thorough and specified accordingly.

Line 128-129: I really cannot understand what is the relationship between "Aubrecht et al. (2015) and Gunasekera et al. (2015)'s approach (although I did read these two papers)" and "the urban/township/rural population proportions of each province in 2010-census data set"; aren't these proportions provided in the 2010-census directly? Or cannot these proportions be easily calculated from info. in the 2010-census directly? Or, did I misunderstand your original intended meaning here? i.e., when talking about "the urban/township/rural population proportion of each province" here, you refer to those of the 2015 GHS data set, right? So, do please try to make the sentence of such kind as accurate as possible, so as to avoid getting

readers lost.

Response: Thank you for pointing this out. We provided a wrong reference of Aubrecht et al. (2015). The right one should be Aubrecht and Leon Torres (2015) with the title "Top-down identification of mixed vs. residential use in urban areas: Evaluation of remotely sensed nighttime lights for a case study in Cuenca City, Ecuador".

In Aubrecht and Leon Torres (2015), they identified the areas of mixed and residential grids geospatially within the urban extent of Cuenca City, Ecuador by using the Impervious Surface Area (ISA) data as the bridge, due to its strong spatial correlation with built-up area. The assumption behind their method was that intense lighting is associated with a high likelihood of commercial and/or industrial presence (which is commonly clustered in certain parts of a city, such as central business districts and/or peripheral commercial zones and such area are defined as "mixed use area") and areas of low light intensity are more likely to be pure residence zone (defined as "residential use area"). The proportions of 25% mixed use area and 75% residential use area were derived from detailed in situ survey data of the Cuenca City. The assignment of the attribute (mixed use, residential use) to the grids in the urban area of the Cuenca City was achieved by sorting the grids according to their ISA intensity value in a max-min order. Then the area of the grids with relatively large ISA intensity value were added up until it reaches the 25% proportion of mixed use area. The cut-off ISA intensity value was derived simultaneously. The remaining grids with smaller intensity value were assigned as "residential use" grid.

In Gunasekera et al. (2015), a similar practice as that in Aubrecht and Leon Torres (2015) was performed in developing the building stock exposure model for the globe. The difference is that Gunasekera et al. (2015) sorted the girds according to the population density in the LandScan population dataset and assigned the gird with urban/rural attribute. The largest and most populated contiguous gridded population data cells are classified as urban. This step was repeated iteratively until the urban population proportion for each country was reached.

In our work, which is similar to that of Gunasekera et al. (2015), the ancillary population density profile we use is the product of the Global Human Settlement project of the European Commission, and the urban/township/rural population proportions for urban/township/rural urbanity are derived from the 2010-census data of China. Then for each province, the grids with largest GHS population are assigned as "urban" until the urban population proportion is reached; the remaining largest population girds are assigned as "township" until the township population proportion is reached; the grids left are assigned as "rural".

Line 130-131: "the population proportion of urban/township/rural urbanity level is 76.64%, 12.66% and 10.7%, respectively", which means that the population proportion of urban/township/rural urbanity levels "in the 2010-census" are 76.64%, 12.66% and 10.7%, respectively", right? If so, please make this info. complete and clear.

Response: Yes. Thank you for pointing this out, we will further specify the description.

Line 131-132: The sentence "Then the grids (1km×1km) in 2015 GHS population density file

of Shanghai are sorted from the largest to the smallest" reads awkward. Is it better if changing it into "Then the grids (1km×1km) of Shanghai in 2015 GHS file are sorted from the largest to the smallest in population density"? Please check.

Response: Accepted. Thank you. The description of "Then the grids (1km×1km) in 2015 GHS population density file of Shanghai are sorted from the largest to the smallest" will be replaced by "Then the grids (1km×1km) of Shanghai in 2015 GHS file are sorted from the largest to the smallest in population density".

Line 128-144: I guess there is an important assumption here, namely, the larger the population density, the higher the urbanized extent. If so, please write this out clearly to avoid making readers have to guess. Please check and revise.

Response: Yes. We will add the following description: "The assumption here is that the higher the grid population density, the higher likelihood it is assigned with 'urban' attribute."

Line 163: "up to now" reads awkward, please check.

Response: Accepted. We will replace "up to now" by "at this stage".

Line 165: "from the 2010-census" reads awkward, please check.

Response: Accepted. We will replace "from the 2010-census" by "from the records in the 2010-census data".

Line 169-170: I don't think "F2" has been explained clearly, including how they were calculated and how they were used subsequently in the amplification of the 2010-census data. I can guess these. But I think it is necessary to introduce them clearly in the text (e.g., using one example), so as to avoid making the readers have to make that guess.

Response: Thank you for pointing this out. We will take Shanghai as an example to show how F2 is derived. In Table 2, we only provided the population for each urbanity level of each province in 2015 GHS population density profile but did not provide the corresponding 2010-census population due to the Table width limitation. But it can be derived from other columns of Table 2. It is equal to the product of "the number of families" and the "average person per family". We will add the above explanation in the revised manuscript.

Line 179-180: Similarly to "F2", the wording for "F3" is also vague and incomplete. Please check and revise.

Response: Accepted. We will also give an example explaining how the value of "F3" is calculated.

Line 183-185: there is "the population in each grid living in building types grouped by number of storey (1, 2-3, 4-6, 7-9, 10) or by construction material (steel/RC, mixed, other, brick/wood) can be derived". It is hard to guess how you achieve this. I guess there is another very important assumption here. Specifically, from the 2010-census, we can get a series of provincial level percentages of the population living in buildings with different floors or with different structure types with one urbanity level (urban, township, or rural); then it is assumed that all the grids with this same urbanity level are all evenly/uniformly have these same percentages. My guess

may be correct, may not. But the author should make this highly generalized statement clear enough, so as to make this key modeling step easier to be understood.

Response: In the 2010-census, for each urbanity level of each province, the whole number of families living in buildings grouped either by storey class or by structure types are recorded. By multiplying the average person per family (which is also given in 2010-census, as listed in Table 2), the whole population within each urbanity level of each province living in different building types can be derived. In this comment, the description of "it is assumed that all the grids with this same urbanity level are all evenly/uniformly have these same percentages" in this comment is not correct. The grids within the same urbanity level are not of the same percentage, but with different apportion weight, which is derived according the population density in the 2015 GHS population ancillary profile. According to this weight, the whole population living in different building types are further disaggregated into each grid. We will add the above explanation to the beginning of this paragraph.

Line 186: "the number of buildings" reads awkward relative to the main topic of this paper. Please check and revise.

Response: Accepted. We will replace "the number of buildings" by "the floor area of buildings".

Line 201: "currently" and "for instance" both read awkward. Please check and revise.

Response: Accepted. We will replace "currently" by "at this stage" and remove "for instance".

Line 205-206: there is "9 equations". My wonder includes what are they and how they function specifically? Please explain them in detail. There is the phrase of "linear problem". why you suddenly say this, why the problem is linear, how this linear problem looks like?

Response: Thank you for pointing this out. The 9 equations are as follows:

$$BRIWOMC1 + STLRCMC1 + MIXEDMC1 + OTHERMC1 = Num_{sotrey1}$$
$$BRIWOMC23 + STLRCMC23 + MIXEDMC23 + OTHERMC23 = Num_{sotrey23}$$
$$STLRCMC46 + MIXEDMC46 + OTHERMC46 = Num_{sotrey46}$$
$$STLRCMC79 + MIXEDMC79 + OTHERMC79 = Num_{sotrey79}$$
$$STLRCMC10 + MIXEDMC10 + OTHERMC10 = Num_{sotrey10}$$
$$BRIWOMC1 + BRIWOMC23 = Num_{BRIWO}$$
$$STLRCMC1 + STLRCMC23 + STLRCMC46 + STLRCMC79 + STLRCMC10 = Num_{STLRC}$$
$$MIXEDMC1 + MIXEDMC23 + MIXEDMC46 + MIXEDMC79 + MIXEDMC10 = Num_{MIXED}$$
$$OTHERMC1 + OTHERMC23 + OTHERMC46 + OTHERMC79 + OTHERMC10 = Num_{OTHER}$$

The to-be-solved variables in the left side of the equation group represent the number of populations living in the 17 sub-types of buildings (as defined in Table 3); on the right side of the equations, the whole number of population living in buildings classified by storey class or by structure type are given for each urbanity level of each province in the 2010-census.

Since this equation group has more unknown variables (17) than the number of equations (9), it is called as underdetermined linear problem. We will add the above explanation to the context.

Line 210-223: this part is very hard to follow. Please check and provide necessary details and explanations. For example, (1) in Line 216-219, there is "the remaining steel/RC buildings are proportioned to other storey classes from highest to lowest", and the like. Please specify, how you get these proportions? (2) regarding step 6 here, I think it is simply that the remaining buildings in each storey class are all belonging to "mixed" buildings.

Response: Thank you for pointing this out. We will give a more detailed and accurate description of the equation solution process.

Line 226: this sub-title cannot meet with the contents below.

Response: Accepted. This sub-title will be changed to "Derive the floor area of the 17 building sub-types".

Line 253-368: i.e., the two validations. Specific comments, please see the second point in the general comments section above.

Response: Thank you. Detailed responses were given to the general comments above.

Line 306-307: "on the other hand" reads awkward. Please check and revise. However?

Response: Accepted. We will remove the description of "on the other hand".

Line 330: I feel more information regarding "UCC" should be provided, so that, it is easier for readers to understand why using UCC can make that adjustment. (existing studies of other researchers might have discussed this, but in the interest of the common requirement that a single paper had better be self-standing, key info. should be introduced)

Response: Thanks for pointing this out. We will provide more explanation regarding the use of UCC.

Line 347-350: I feel the explanation of this "de-amplification factor" is not clear enough, more info. is needed. For example, I found that "1.32" is exactly the arithmetic mean of "1.33, 1.34 and 1.29". Is this just a coincidence? Please check and revise.

Response: Thank you for this comment. For each province, the de-amplification factor is equal to the sum of the product between the amplification factor of each urbanity level and its floor area ratio. For example, for Shanghai, the de-amplification ratio of 1.32 is calculated as follows: $1.33 \times 63\% + 1.34 \times 15\% + 1.29 \times 22\% = 1.3227 \approx 1.32$ (these numbers are given in Table 6). It is a coincidence that $(1.33+1.34+1.29)/3=1.32$. We will add the above example to the context to better illustrate the derivation process of the de-amplification factor.

Line 414-420: I feel the current limitation discussion is too general. Instead, I feel the most relevant and direct limitation discussion (disadvantages and future improvement directions) should focus on those assumptions and "factors" that this modeling process used.

Response: Thank you for this suggestion. We will strengthen the discussion on limitations related to the assumptions and approximations used in this work.

**References**

Aubrecht, C. and Leon Torres, J. A.: Top-down identification of mixed vs. residential use in urban areas: Evaluation of remotely sensed nighttime lights for a case study in Cuenca City, Ecuador., 2015.

Aubrecht, C., Gunasekera, R., Ishizawa, O. and Pita, G.: The flipside of "urban"—A novel model for built-up-adjusted rural-urban pattern identification and population reallocation, in CDRP Working Paper (Country Disaster Risk Profiles Initiative), The World Bank-Social, Urban, Rural & Resilience (GSURR), Disaster Risk Management (DRM) Washington DC, USA., 2015.

Chen, Z., Li, Z., Ding, W. and Han, Z.: Study of Spatial Population Distribution in Earthquake Disaster Reduction ---- A Case Study of 2007 Ning'er Earthquake, Technology for Earthquake Disaster Prevention, 7(3), 273–284, 2012.

Gunasekera, R., Ishizawa, O., Aubrecht, C., Blankespoor, B., Murray, S., Pomonis, A. and Daniell, J.: Developing an adaptive global exposure model to support the generation of country disaster risk profiles, Earth-Science Reviews, 150, 594–608, 2015.

Han, Z., Li, Z., Chen, Z., Ding, W. and Wang, L.: Population, Housing Statistics Data Spatialization Research in the Application of Rapid Earthquake Loss Assessment ---- A Case of Yiliang Earthquake, Seismology and Geology, 35(4), 894–906, doi:10.3969/j.issn.0253-4967.2013.04.018, 2013.

Wu, J., Li, N. and SHI, P.: Benchmark wealth capital stock estimations across China's 344 prefectures: 1978 to 2012, China Economic Review, 31, 288–302, doi:10.1016/j.chieco.2014.10.008, 2014.

Wu, J., Ye, M., Wang, X. and Koks, E.: Building Asset Value Mapping in Support of Flood Risk Assessments: A Case Study of Shanghai, China, Sustainability, 11(4), 971, doi:10.3390/su11040971, 2019.

Xu, J., An, J. and Nie, G.: A quick earthquake disaster loss assessment method supported by dasymetric data for emergency response in China, Nat. Hazards Earth Syst. Sci, 16, 885–99, 2016.

---

## Author Comment (AC2) · 30 Jul 2020

Comment 1: The Introduction starts with some common sentences on earthquake loss and related action on the UN level, however, this paragraph is not very suitable to introduce the topic of the manuscript to the potential readers. Therefore, I kindly recommend rewriting.

Response: Thank you for this suggestion. We will reorganize this part in the revised manuscript by using materials more closely related to our focus.

Comment 2: Moreover, the statement of world-wide earthquake loss should go clearly beyond the referred two studies of one of the co-authors of this manuscript.

Response: Thank you. We will add other references to this statement.

Comment 3: One more detail: For me it is not clear why in a paper from 2011 earthquake loss can be given in 2016 values, please clarify.

Response: This is because the work in Daniell et al. (2017) is based on the work of Daniell et al. (2011). To avoid confusion, we will delete the latter and only keep the former reference.

Comment 4: Then the authors address the need to have information on the building stock level when risk assessment should be undertaken. They state that in cases where such information is not available, obtaining necessary information is not practicable. I strongly disagree with this statement, throughout the relevant literature there are many different methods presented of how to do so. This needs thorough revision.

Response: Thank you for this comment. We will replace the description of "obtaining necessary information is not practicable" by "a series of ancillary data based on remote sensing technologies will be resorted to". And a succinct summarization of related methods will be added.

Comment 5: Moreover, the authors elaborate on a method to compile such information by taking census data in consideration. How the building stock value is correlating to statistical information on population density (lines 49-57)? How did the authors generally treat the MAUP issue when using data bond to administrative borders?

Response: A detailed description of the data used and the modelling process we adopted are provided in the "Data Sources and Methodology" section. In short, the building stock value (here we use the replacement value) is derived by multiplying the residential building floor area and the unit construction price we compiled for the building types used in this work. And the residential building floor area is derived directly from the statistical information provided in the 2010-census of China. Since the 2010-cenus data are categorized at urbanity level, to disaggregate the census records into grid level, the 2015 Global Human Settlement population density profile is used to derive the apportion weight of each grid.

If MAUP refers to the "modifiable areal unit problem", then we did not consider the MAUP problem during the modeling process. According to the explanation of MAUP from the internet: "MAUP is a statistical biasing effect when samples in a given area are used to represent information such as density in a given area. The area defined by an analyst is often arbitrary,

thus measurement such as density could be deceptive because that density measure could have widely different results based on shape and scale chosen for analysis." However, in this work the provincial boundaries are not arbitrarily defined and the 2010-census data to be disaggregated is also derived from survey population within each province administratively. Therefore, all the boundaries are clearly defined. What we need to do is to distribute the census data into grid level for each province. Therefore, we do not think the MAUP issue will affect our modelled results and to be honest, we have not seen discussions on this MAUP issue in studies of this kind even after an extended literature review.

Comment 6: Further down the text body, the authors correctly state that "to better cope with this spatial mismatch between natural hazards [spatial occurrence] and administrative boundaries, building stock models should be geocoded with relatively high resolution and be disaggregated from more detailed census data". The last statement means that from a methodological point of view, such an assessment will not guide us to precise results that can be used as a proxy for the building stock. So somehow, the introductory section is unclear with respect to what the authors would like to show us in their study.

Response: Thank you for pointing this out. You are right that geocoded resolution referred to in this study (1km*1km) is still not specified enough to provide precise results that can be used as a proxy for the building stock. The purpose of this work is to develop grid-level residential building stock model for seismic risk assessment in mainland China and this is achieved by disaggregating 2010-census into 1km*1km grid with the 2015 GHS population density profile as the bridge. We will add more method description in the introduction part to avoid possible confusion and misunderstanding.

Comment 7: Finally, the research gap is not properly defined, nor is the niche to be filled by this work easily accessible to potential readers. In the method section the authors explain how they aggregated information on the building characteristics to information on the population density, both at a final resolution of 1 km grid cells. In this respect it remains unclear how the other building types were excluded from the grids, as information on e.g. building design and material in the statistics ("Long Table") are also related to other building types, right? -> Needs clarification.

Response: Thank you for this comment. No building type is excluded in the modelling process. The 2010-census recorded number of families living in buildings classified by four structure type (brick-wood, mixed, steel-RC, other) and by five story classes (1, 2-3, 4-6, 7-9, ≥10) is regrouped into 17 building sub-types attached with structure type and story class. Based on these building sub-types, the unit construction price for each of them will be easier to be compiled.

Comment 8: Further on, the authors present different methods of how to merge different types of information such as the amount of buildings of different height or different construction type to these grid cells, resulting in a certain spatial probability for the different data. It remains open, however, how this information was finally be checked against the real world situation, and as such it remains open. How e.g. information on population was distributed or allocated to different building categories?

Response: For each urbanity level of each province, number of families living in different residential building types (classified by structure type or story) are given in the 2010-census. The average population per family and the average floor area per capita are also given in the 2010-census. Therefore, the overall population living in each building type and the total floor area of different building types (classified by structure type or story) can be derived for each urbanity of each province. This is the way how information on population and building is related. We now need to distribute the overall building floor area for the whole urbanity into grid level, based on the apportion weight derived from the 2015 GHS population density profile.

Comment 9: Occupation rate and building values were then allocated to the different building sub-categories, and spatially distributed over the grid cells. Results of values per grid cell where then compared to (A) a study published by Wu et al. (2014) on the net capital stock, (B) more detailed information available on the residential floor area for the Shanghai district, and (C) an empirical earthquake vulnerability study published by one of the co-authors of this manuscript, linking vulnerability to reported loss. The authors conclude that the results from the present manuscript (in terms of what? Potential value of buildings? Potential loss resulting from an earthquake scenario?) are in line with results from other studies, a statement which cannot be supported by the referee evaluating the information provided by the authors. In the present form, the results of the study are not validated, they are only opposed to other studies on building values (in case A), to the area used for computation of values (in case B) and to vulnerability, linking the newly generated building values to an empirical vulnerability function and comparing the results to some loss reports available (in case C).

Response: We are afraid we cannot agree with the conclusions in the comment here. We believe you also agree that there are different levels of validation tests, from a general comparison of an overall value, to an exact match of data distribution pattern, or even an building-by-building comparison. Ideally, the best validation practice will be comparing the modelled results with in-situ site surveys, which will be undoubtedly expensive and extremely time consuming for we to do this for the whole mainland China, also we cannot afford to do this. It is 35 years ago that the Chinese government conducted its last country-wide building survey in 1985. However, considering the main application of the model developed in this study is for seismic risk assessment, an exact validation may be not necessary. Firstly, when damaging earthquake occurs, it usually affect at least several counties or even provinces. What we want to do is develop a relatively high-resolution building stock model based on the open census data. When distributing the census data into grid level, to better reflect the spatial heterogeneity of population and building distribution, we use the 2015 GHS population density profile as the bridge. And we believe it is better than distribute census data into an administrative unit evenly.

When it comes to the validation of our modelled building stock replacement value and floor area, a reasonable test is to check whether the model can be used to better reflect the seismic loss distribution pattern or to compare modelled results with outputs derived by using different methods at coarser level. And we do not think this is a "self validates self" thing, since when disaggregating the urbanity level census data into grid level, this process is not related to or affected by prefecture or county/district boundary. And we think if we reaggregate the grid-level data into county/district/prefecture level and compare them with other independent

sources of validation data. If they are still consistent, then it can be regarded as a positive signal indicating the reliability of our modelled results. The three validation tests in this work are performed for different purposes.

(A) The comparison with Wu et al. (2014) is to first make sure that the replacement value of residential building stock we modelled is of reasonable magnitude. Since our model and that of Wu et al. (2014) are different in method, model target and the treatment of depreciation issue, but the ratio between our residential building replacement value and the net capital stock value in Wu et al. (2014) is within the range of 0.31~0.65. Recently, for each province we also checked the average ratio between newly constructed residential building value and the value of all newly constructed buildings in the past five years. This ratio is within the range of 0.38~0.75. Through this comparison, we consider that the replacement value of residential building stock we estimate for each province is reasonable.

(B) We then compare our modelled residential building floor area at district level with the statistical results of Shanghai only, since such data in other provinces are not available. Since the disaggregation of the urbanity level census data into grid level is not limited by or related to county/district boundary, when our modelled results are reaggregated into county/district level and have high correlation coefficient (0.91 without adjustment) with county-level statistical records, we believe this consistency is suggesting the reliability of our modelled residential building floor area for Shanghai.

Recently, after a broader check of databases related to building statistical data, we found that although in other provinces there are no such records of accumulated residential building floor area like that given in Shanghai 2015 statistical yearbook, there are provincial 2010-cenus data available (although not with open access, but can be bought), in which the residential building floor area can be directly derived for each county/district. We will allocate these data and further validate our modelled residential floor area for all counties/districts of all the 31 provinces. The comparison will be added to the revised manuscript.

(C) In applying our modelled residential building replacement value to loss estimation by using the intensity map of Wenchuan Ms8.0 earthquake, we are trying to check the availability of our modelled results in risk analysis, since the model is targeted for seismic risk analysis.

We do not think that the empirical loss curve derived by Daniell (2014) is used in this process is a problem. First of all, this empirical loss curve was derived long before the preparation of this study. Secondly, the derivation of this empirical curve is based on extensive collection of damage and loss records from journals, books, reports, conference proceedings and even newspapers. Therefore, only because the author of this empirical curve is also a co-author of this work should not decrease the reliability of his previous research findings. In Daniell (2014), he also developed empirical loss curves for other countries and regions through careful selection and rectification of historical records and it turns out that his curve performs better than previous studies in predicting seismic loss, since he considered the change in vulnerability of exposure with time. That is why in the study, the empirical loss curve in Daniell (2014) for

mainland China is used.

Since our estimated loss is around 144-288 billion RMB (in 2015 current price), while the median loss estimated from post-earthquake investigation is around 212.32-247.25 billion RMB (in 2008 current price) for residential buildings in Sichuan province, we think this can be taken as a consistent result. Of course.

Comment 10: With respect to the latter, further questions arise with respect to different construction types and their individual structural vulnerability concerning earthquakes, this needs careful interpretation and more information on the comparison performed.

Response: In case C, the empirical loss curve of Daniell (2014) gives the relation between macroseismic intensity and loss ratio (the ratio between building repairment cost and replacement cost). This curve is called as "empirical" because it is regressed from the loss ratio and intensity pairs of historical damaging earthquakes. And the loss ratio is the derived from the loss and exposed value considering all building types. Therefore, there is no further classification of buildings when applying this empirical curve in rapid loss estimation, only the seismic intensity map and the sum of the replacement value of all buildings in each grid are needed.

It is worth to note that in this work, the replacement value not the repairment value is used in the loss estimation, which means the depreciation of buildings are not considered. The difference in building vulnerability will not affect the estimated replacement value, but will affect the repairment value greatly.

Comment 11: As such, the added value of the material presented here is not clear to me. Statements such as "Therefore, the estimated loss range, based on the buildings stock model developed in this study and the empirical loss function developed in Daniell (2014), is quite compatible with that given in previous studies. This compatibility further validates the robustness of our residential building stock model" (lines 392ff.) may be right but cannot be proven from what the authors have shown.

Response: We fully understand your concern, since the loss estimation process involves uncertainties from both the estimation of replacement value of residential buildings and the uncertainty in development of the empirical loss curve. That is why the estimated loss is given in range form, not a specific value. The reliability of the modelled replacement value and floor area have been checked in case A and case B. In case C, we want to further check the prospect of applying the modelled results in seismic risk assessment and our estimated loss of 144-288 billion RMB (in 2015 current price) is approximate to loss estimated from post-earthquake investigation, which is 212.32-247.25 billion RMB (in 2008 current price). We think this compatibility indicates the positive prospect of applying our model to seismic risk assessment.

Comment 12: Finally, information given in the conclusion is by far too generalized, instead, the overall limitation and the specific ones, such as those arising from the use of aggregated statistical data and the underlying factors, should be discussed in detail.

Response: Thank you for your patience to read this manuscript through. We will strengthen

discussion on limitations of this work.

Comment 13: Hence, I cannot recommend publication of the work in its present form, the manuscript needs major revisions in terms of (1) problem statement, including extended literature review, (2) explanation of methods, (3) compilation of results, and – most important –(4) discussion, including discussion on limitations and uncertainties, so that (5) sound conclusions will become possible.
Response: Thank you for your careful review. We will take these comments seriously and modify our manuscript accordingly in the revised version.

**References:**

Daniell, J.: Development of socio-economic fragility functions for use in worldwide rapid earthquake loss estimation procedures, Ph.D. Thesis, Karlsruhe Institute of Technology, Karlsruhe, Germany., 2014.

Daniell, J. E., Khazai, B., Wenzel, F. and Vervaeck, A.: The CATDAT damaging earthquakes database, Natural hazards and earth system sciences, 11(8), 2235–2251, doi:https://doi.org/10.5194/nhess-11-2235-2011, 2011.

Daniell, J. E., Schaefer, A. M. and Wenzel, F.: Losses Associated with Secondary Effects in Earthquakes, Front. Built Environ., 3, doi:10.3389/fbuil.2017.00030, 2017.

Wu, J., Li, N. and SHI, P.: Benchmark wealth capital stock estimations across China's 344 prefectures: 1978 to 2012, China Economic Review, 31, 288–302, doi:10.1016/j.chieco.2014.10.008, 2014.